# Multiple Object Stitching for Unsupervised Representation Learning

## Abstract

Contrastive learning for single object centric images has achieved remarkable progress on unsupervised representation, but suffering inferior performance on the widespread images with multiple objects. In this paper, we propose a simple but effective method, *Multiple Object Stitching* (*MOS*), to refine the unsupervised representation for multi-object images. Specifically, we construct the multi-object images by stitching the single object centric ones, where the objects in the synthesized multi-object images are predetermined. Hence, compared to the existing contrastive methods, our method provides additional object correspondences between multi-object images without human annotations. In this manner, our method pays more attention to the representations of each object in multi-object image, thus providing more detailed representations for complicated downstream tasks, such as object detection and semantic segmentation. Experimental results on ImageNet, CIFAR and COCO datasets demonstrate that our proposed method achieves the leading unsupervised representation performance on both single object centric images and multi-object ones. The source code can be found in the supplementary material.

## 1 Introduction

Contrastive learning methods (Wu et al., 2018; Ye et al., 2019; Chen et al., 2020a; He et al., 2020; Chen et al., 2021) have witnessed impressive development on unsupervised visual representation learning. In spite of massive designs (Chen et al., 2020a; He et al., 2020; Grill et al., 2020) proposed to improve performance of contrastive learning, these methods can be summarized into an instance discrimination task. In this pretext task, the unlabeled images are augmented into several views using different data augmentations, where the distorted views from the one image are regarded as the samples of the same "instance class". The contrastive objective is to discriminate the distorted views of the same image instance from other instances, called instance discrimination. This contrastive configuration works well on the single object centric datasets, which mainly contains images with a single object as the subject, such as ImageNet (Russakovsky et al., 2015). However, the model pretrained on single object centric images is optimized to represent the image in a global manner, thus prone to miss local single object features in multi-object images, then leading to degenerated performance on multi-object images.

An intuitive solution to tackle the above problem is to conduct contrastive learning on multi-object images. However, directly applying image-level contrastive learning on multi-object images suffers a semantics inconsistency issue. As shown in Figure 1 (a), since multiple objects are randomly distributed in the image, the views obtained by the random crop strategy may contain significantly different objects, namely *semantics inconsistency issue* in multi-object images. These incompatible views introduce false positive pairs for image-level contrast in multi-object images and mislead the training of model, then leading to inferior representation quality.

To alleviate the above semantics inconsistency issue, a straightforward method is to implement contrastive learning in the object level, as shown in Figure 1 (b). However, the access to objects in the multi-object image requires additional human annotations, which doesn't meet the settings of unsupervised representation learning. To this end, region-level (Yang et al., 2021; Roh et al., 2021; Li et al., 2022; Wei et al., 2021; Xie et al., 2021b; Xiao et al., 2021; Yun et al., 2022; Zhang et al., 2023) and pixel-level (Liu et al., 2020; Xie et al., 2021c; O. Pinheiro et al., 2020; Wang et al., 2021;

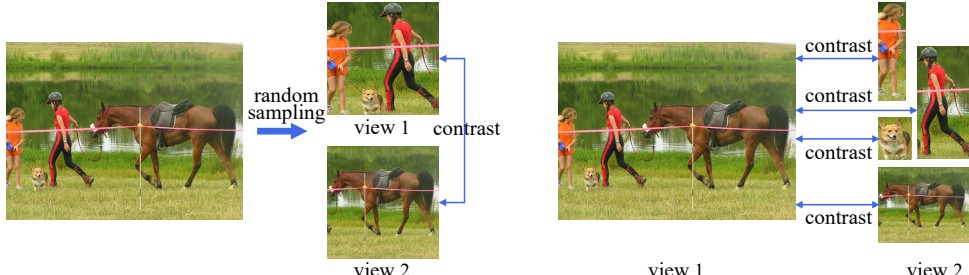

(a) **image-level contrast**: potential false positive pair     (b) **object-level contrast**: multi-objects *vs* single objects

Figure 1: Image-level contrast *vs* object-level contrast. (a) Image-level contrast takes random crops from one image as positive pair, which introduces potential false positive pair; (b) Object-level contrast constructs positive pairs between the multi-object image and objects in it, respectively, which provides more accurate targets for contrastive learning.

Islam et al., 2023) contrastive learning methods are proposed. These methods conduct contrastive learning according to the correspondences between regions or pixel from different views and focus on the representations of parts from image, instead of the global ones of image, thus alleviating the semantics inconsistency issue. Moreover, saliency-based contrastive learning methods (Selvaraju et al., 2021; Mo et al., 2021; Xie et al., 2021b) leverage class activation map to construct object correspondences between positive views. In spite of promising improvement over plain contrastive learning, the above methods fail to provide more accurate object-level correspondences.

In this paper, we further propose a simple but effective method, *Multiple Object Stitching* (*MOS*) strategy, for object-level contrast in an unsupervised manner. Specifically, we stitch several single object images into an artificial multi-object image, to construct object-level correspondences between image views without human annotations. Since Vision Transformer architecture (Dosovitskiy et al., 2021) (ViT) is less sensitive to artificiality produced by the boundary of image stitching, the proposed strategy can effectively simulate multi-object scenarios in reality. By contrasting the synthesized multi-object image and the corresponding single object images, the trained model is encouraged to represent all objects in the image and preserves richer features for various complicated downstream tasks.

To effectively discriminate the objects from each others in the multi-object scenarios, we implement contrastive learning as the following aspects. **(I)** The contrast between multi-object image and the corresponding single object views guides the model to discriminate each object in the image. **(II)** The contrast between multi-object image and another multi-object view constructs more complicated object correspondences to refine the representations of multi-object images. **(III)** The contrast between single object image and another single object image view is used to eliminate the potential representation gap produced by the domain gap between the stitched images and the natural ones.

Experimental results on ViT-S backbone demonstrate that our method reaches 83.5% finetune accuracy, 77.9% linear accuracy and 74.2% kNN accuracy on ImageNet-1K, 98.3% finetune accuracy, 96.3% linear accuracy and 95.1% kNN accuracy on CIFAR10, 86.1% finetune accuracy, 78.5% linear accuracy and 73.5% kNN accuracy on CIFAR100, 45.6 $AP^{bb}$ and 40.6 $AP^{mk}$ on COCO, significantly outperforming the previous state-of-the-art methods on both single object centric images and multi-object ones.

Our main contributions can be summarized as follows: **(I)** We introduce an effective object-level contrast strategy, *Multiple Object Stitching*, for unsupervised representation learning of multi-object images, which significantly alleviates *semantics inconsistency issue* suffered by its contrastive counterparts. **(II)** Three dedicated contrastive objectives are proposed to model multiple-to-single, multiple-to-multiple and single-to-single representations for more complicated downstream tasks, e.g. object detection and instance segmentation. **(III)** The experimental results demonstrate that our proposed method achieves state-of-the-art performance on both single object centric images (ImageNet-1K and CIFAR) and multi-object ones (COCO).

## 2 RELATED WORK

In this section, we discuss the related contrastive learning works, including contrastive learning for single object centric representations and the one for multiple object representations.

### 2.1 CONTRASTIVE LEARNING FOR SINGLE OBJECT CENTRIC REPRESENTATIONS

Under the assumption of image semantics consistency with various image augmentations, contrastive learning methods (Wu et al., 2018; Ye et al., 2019; Chen et al., 2020a) optimize the neural networks by discriminating image instance under different data augmentations from other image instances, namely instance discrimination task. The representative works can be summarized as follows. SimCLR (Chen et al., 2020a) empirically presents several key points for contrastive learning, more data augmentations, learnable nonlinear module, larger batch size and longer training schedule. MoCo series methods (He et al., 2020; Chen et al., 2021) introduce momentum encoder to keep the consistency of keys during pretraining and effectively improve the performance. BYOL (Grill et al., 2020) and Simsiam (Chen & He, 2021) explore the contrastive framework without negative pairs. SwAV (Caron et al., 2020) and DINO (Caron et al., 2021) propose clustering based contrastive method, where the output features are contrasted with a series of parameterized clusters. Barlow Twins (Zbontar et al., 2021) applies redundancy reduction objective to conduct unsupervised representation learning. MosRep (Wang et al., 2023) randomly composes different images into a mosaic view to enrich the contextual background for contrastive learning of single object centric images. Compared to MosRep, the main difference is that our method regards the stitched image as a whole and then discriminates different objects according to the proposed object correspondence targets. Therefore, our method can better adapt to the representation learning of multi-object images.

Nevertheless, semantics consistency assumption only works well on single object centric images. For multi-object images, the key augmentation technique, random crop operation is prone to cause semantics inconsistency among different crop views, due to potential significantly different objects in difference views. Therefore, the above methods are not directly applicable to more complex multi-object images. In contrast, our proposed method constructs object correspondences by synthesizing multi-object images from single object centric images, where each objects in multi-object image can distinctly correspond to single object centric images, thus eliminating the above semantics inconsistency issue.

### 2.2 CONTRASTIVE LEARNING FOR MULTIPLE OBJECT REPRESENTATIONS

To address the semantics consistency issue in multi-object images, contrastive learning methods for multiple object representation are proposed, including region-based and pixel-based methods. Region-based methods conduct contrastive learning on the region proposals sampled from the original image. InsLoc (Yang et al., 2021) constructs instance localization task by embedding target foreground object into background images. SCRL (Roh et al., 2021) learns invariant spatial representation by contrasting the same cropped region under two augmented views of the given image. ReSim (Xiao et al., 2021) aligns the features in the corresponding sliding windows from the overlapping area between two views of the given image. SoCo (Wei et al., 2021), ORC (Xie et al., 2021b) and UniVIP (Li et al., 2022) obtain object proposals by selective search strategy (Uijlings et al., 2013) and then conduct contrastive learning between the above proposals. DetCo (Xie et al., 2021a) learns local sensitive representations by imposing contrastive learning between global image and local patches with multi-level supervision. SelfPatch (Yun et al., 2022) and ADCLR (Zhang et al., 2023) explores patch-level contrastive learning on Vision Transformer architecture.

Pixel-based methods adopt feature-level pixel alignment using contrastive learning to learn local sensitive representations. Self-EMD (Liu et al., 2020) applies discrete Earth Mover's Distance to compute the spatial similarity among all location pairs on the feature space of two views. VADeR (O. Pinheiro et al., 2020) adopts encoder-decoder architecture to learn dense representations by contrastive learning between pixel pair in representation space. PixPro (Xie et al., 2021c) introduces pixel-propagation module to maximize the similarity between pixel and the corresponding propagating features of pixels. DenseCL (Wang et al., 2021) aligns pixel-wise features according to pixel correspondences computed by feature similarities. LCL (Islam et al., 2023) conducts pixel-level contrastive learning by pixel correspondences cross augmented views.

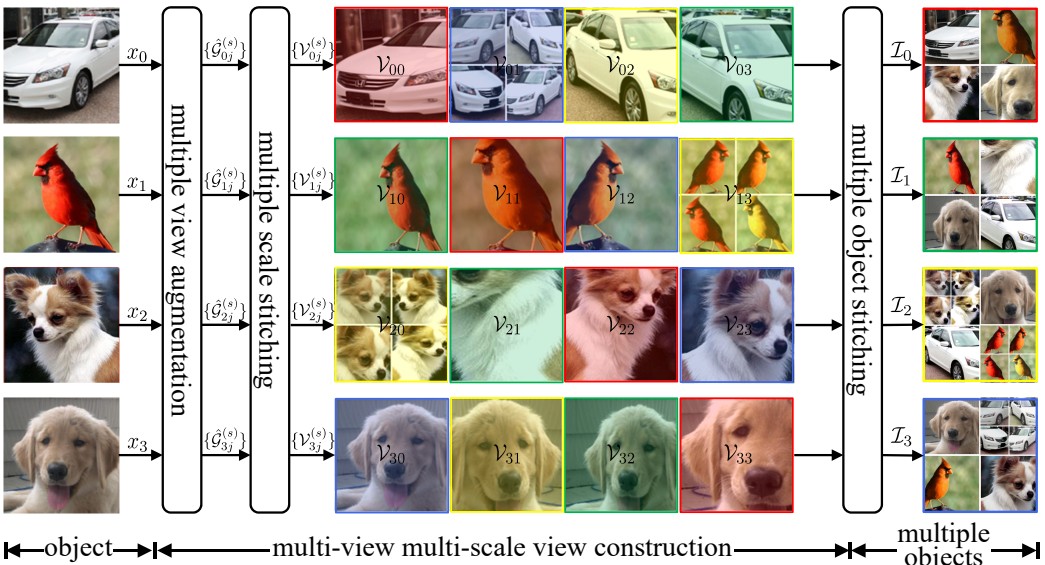

Figure 2: Multiple object stitching workflow. First, for better diversity, the input image $x_i$ is augmented into a series of views $\{\hat{x}_{ij}^{(s)}\}_{j=0}^{(s \cdot r)^2 - 1}$ and then stitched into scaled views $\{\hat{G}_{ij}^{(s)}\}_{j=0}^{r^2-1}$. Second, we randomly sample $\{\hat{G}_{ij}^{(s)}\}_{j=0}^{r^2-1}$ with different scale factors $s$ to obtain multi-scale views $\{\mathcal{V}_{ij}\}_{j=0}^{r^2-1}$. Third, multi-scale views $\{\mathcal{V}_{ij}\}_{j=0}^{r^2-1}$ are stitched into a multi-scale multi-object image $\mathcal{I}_i$.

Saliency-based methods exploit class activation map to discriminate different objects, thus achieving object-level contrastive learning of multi-object images. ContraCAM (Mo et al., 2021) and CAST (Selvaraju et al., 2021) adopt contrastive class activation map to conduct object-aware random crop for multi-object contrastive learning. ORL (Xie et al., 2021b) obtains object-level correspondences by leveraging image-level contrastive pretraining as prior.

In spite of encouraging performance achieved, the above methods heavily rely on specific region proposal strategy, where potential incorrect proposals may hurt the representation performance. Moreover, their performance on single object centric images is prone to degenerate. In this paper, we simulate multi-object scenarios by combining off-the-shelf single object centric images and establish clear visual correspondences between multiple views, thus providing more accurate supervision to learn multi-object representations. Meanwhile, compared to the above methods, our method can keep very competitive performance on single object centric images.

Moreover, OAC (Mishra et al., 2022) proposes an object-aware cropping strategy by dilating the cropped region based on the first views. This method can easily construct object-level correspondences, but also limits the scale variation among positive views.

## 3 THE PROPOSED METHOD

### 3.1 MULTIPLE OBJECT STITCHING

To simulate multi-object scenarios in natural images, we construct multi-object images by stitching off-the-shelf single object centric images and generate the training targets according to the correspondences between different views. In this manner, we can easily determine the context of multi-object images without human annotations, which guides the train model to learn multi-object-aware representations. The workflow of multi-object image stitching is presented in Figure 4.

Let $x_i$ denotes the $i$-th sample in the image batch $x = \{x_i\}_{i=0}^{N-1}$ [1], where $x_i \in \mathbb{R}^{H \times W \times C}$ and $N$ is batch size. To diversify the synthesized multi-object views and accommodate multiple objects in

---

[1]In this paper, "$\{\cdot\}$" denotes ordered sequence, instead of unordered set.

one image without the increment of image size, we apply popular random data augmentations on $x_i$ to obtain $r^2$ small views, written as $\{\hat{x}_{ij}\}_{j=0}^{r^2-1} = \mathcal{T}(x_i, r, 1)$, where each view is resized into $\frac{1}{r}$ size of $x_i$, namely $\hat{x}_{ij} \in \mathbb{R}^{\frac{H}{r} \times \frac{W}{r} \times C}$.

To further simulate multi-scale multi-object scenarios in nature, we synthesize multiple smaller objects in a single view by introducing additional scale factor $s$ into data transform function as $\mathcal{T}(x_i, r, s)$. Specifically, the image sample $x_i$ is augmented into a series of views by

$$\{\hat{x}_{ij}^{(s)}\}_{j=0}^{(s\cdot r)^2-1} = \mathcal{T}(x_i, r, s), \tag{1}$$

where $\hat{x}_{ij}^{(s)} \in \mathbb{R}^{\frac{H}{s\cdot r} \times \frac{W}{s\cdot r} \times C}$. To fit the image size of $\hat{x}_{ij}^{(s)}$ to $\hat{x}_{ij}$, we divide the view sequence $\{\hat{x}_{ij}^{(s)}\}_{j=0}^{(s\cdot r)^2-1}$ into $r^2$ groups, where each group can be presented as $G_{ij}^{(s)} = \{\hat{x}_{ik}^{(s)}\}_{k=j\cdot s^2}^{(j+1)\cdot s^2-1}$. Then the image views in group $G_{ij}^{(s)}$ is stitched into a scaled image by

$$\hat{G}_{ij}^{(s)} = \text{stitch}\,(G_{ij}^{(s)}, s), \tag{2}$$

where $G_{ij}^{(s)} \in \mathbb{R}^{s^2 \times \frac{H}{r} \times \frac{W}{r} \times C}$ is transformed into $\hat{G}_{ij}^{(s)} \in \mathbb{R}^{s\frac{H}{r} \times s\frac{W}{r} \times C}$. For brevity, the multi-view sequence can be written as

$$\{\hat{G}_{ij}^{(s)}\}_{j=0}^{r^2-1} = \mathcal{T}(x_i, r, s). \tag{3}$$

Moreover, when $s = 1$, $\hat{G}_{ij}^{(s)} = \hat{x}_{ij}$.

To simulate wide range of object scale in natural images, we randomly sample scale factor $s \in \{1, 2, \cdots, S\}$ to obtain views with various scales by

$$\mathcal{V}_{ij} = \text{random\_sample}\,(\hat{G}_{ij}^{(s)}). \tag{4}$$

In this way, we can obtain image batch $\{\{\mathcal{V}_{ij}\}_{j=0}^{r^2-1}\}_{i=0}^{N-1}$, where each sample contains $r^2$ views.

Afterwards, we construct multi-scale multi-object images by mixing the above views $\{\{\mathcal{V}_{ij}\}_{j=0}^{r^2-1}\}_{i=0}^{N-1}$. Specifically, we rearrange the views from different samples to form multi-scale multi-object image $\mathcal{I}_i$ by

$$\mathcal{I}_i = \text{stitch}\,(\{\mathcal{V}_{u(i,j)j}\}_{j=0}^{r^2-1}, r), \tag{5}$$

where $u(i, j) = (i + j) \bmod N$ [2] denotes the sample index in image batch for multiple object stitching. In summary, each synthesized multi-object image contains $r^2$ objects, each of which includes $s^2$ views.

For efficient tensor implementation of multi-object stitching, we reform the above formula as tensor form. Specifically, we flatten the indices of object views $\{\{\mathcal{V}_{ij}\}_{j=0}^{r^2-1}\}_{i=0}^{N-1}$ in image batch as

$$t = \{\{i \cdot r^2 + j\}_{j=0}^{r^2-1}\}_{i=0}^{N-1}. \tag{6}$$

The indices for image stitching can be rewritten as

$$q = (t + (t \bmod r^2) \cdot r^2) \bmod T, \text{[2]} \tag{7}$$

where $T = N \cdot r^2$. Hence, the multi-scale multi-object stitching strategy can be presented as $\mathcal{I} = \text{index}\,(\mathcal{V}, q)$, where $\text{index}\,(\cdot, \cdot)$ rearranges the elements in the sequence by the given indices. The labels for multi-object image batch $\mathcal{I}$ with respect to single-object image can be obtained by

$$y^{\text{m2s}} = \{\{u(i,j)\}_{j=0}^{r^2-1}\}_{i=0}^{N-1}. \tag{8}$$

The labels for the contrast between multi-object image batch and multi-object image batch can be presented as

$$y^{\text{m2m}} = \{\{l\}_{l=(i-r^2+1+N) \bmod N}^{(i+r^2-1) \bmod N}\}_{i=0}^{N-1}, \text{[2]} \tag{9}$$

which means that one multi-object image corresponds to other $(2r^2 - 1)$ multi-object images, due to the object overlapping. Furthermore, due to the different number of object overlapping between multi-object images, the similarity scores for each item in label $y^{\text{m2m}}$ can be written as

$$\omega^{\text{m2m}} = \{\{1 - |r^2 - l - 1|/r^2\}_{l=0}^{2r^2-2}\}_{i=0}^{N-1}. \text{[2]} \tag{10}$$

---

[2] More explanations can be found in the appendix.

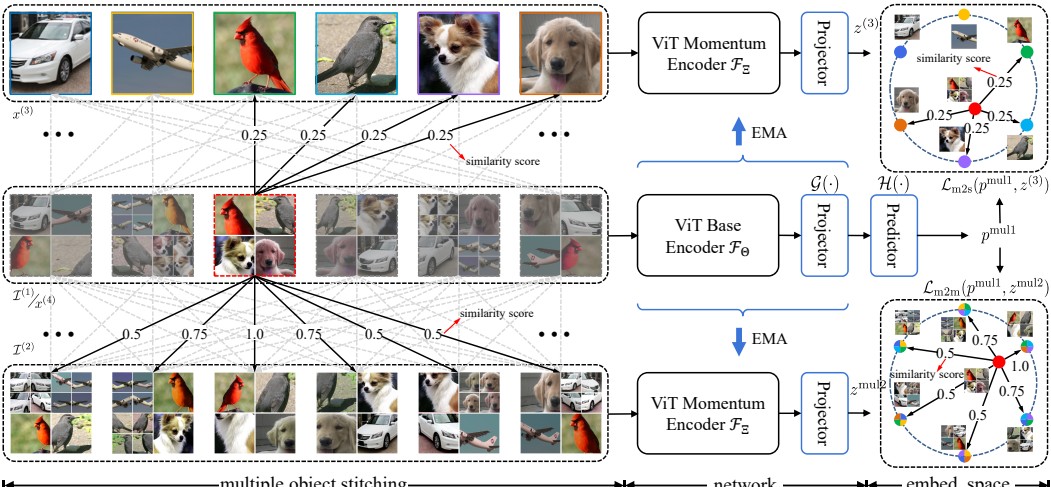

Figure 3: The framework of multiple object correspondence contrast. The proposed multi-object stitching strategy is applied to the input image batch to construct the similarity scores between synthesized multi-object images and single object centric ones. Then, ViT model is trained by both multiple-to-single and multiple-to-multiple contrastive objective.

## 3.2 MULTIPLE OBJECT CORRESPONDENCE CONTRAST

In this section, we apply the synthesized multi-object images and their correspondences to guide the trained model to capture multi-object-aware representations in natural images.

As shown in Figure 3, the input image batch $x$ is transformed into 4 views, including two multi-object views ($\mathcal{I}^{(1)}$ and $\mathcal{I}^{(2)}$) obtained by multiple object stitching strategy and two single-object views ($x^{(3)}$ and $x^{(4)}$) obtained by plain data augmentations. To implement contrastive learning, we adopt base-momentum encoder framework as MoCo-v3 (Chen et al., 2021), which includes base encoder $\mathcal{F}_\Theta$, momentum encoder $\mathcal{F}_\Xi$, projection module $\mathcal{G}$ and prediction module $\mathcal{H}$. The multi-object image batch $\mathcal{I}^{(1)}$ and single object image batch $x^{(3)}$ are successively fed into base encoder $\mathcal{F}_\Theta$, projector $\mathcal{G}$ and predictor $\mathcal{H}$ to obtain the prediction representation $p^{\mathrm{mul1}} = \mathcal{H}(\mathcal{G}(\mathcal{F}_\Theta(\mathcal{I}^{(1)})))$ and $p^{(3)} = \mathcal{H}(\mathcal{G}(\mathcal{F}_\Theta(x^{(3)})))$. Then, image batch $\mathcal{I}^{(2)}$, $x^{(3)}$ and $x^{(4)}$ are successively fed into momentum encoder $\mathcal{F}_\Xi$ and projector $\mathcal{G}$ to obtain projection representation $z^{\mathrm{mul2}} = \mathcal{G}(\mathcal{F}_\Xi(\mathcal{I}^{(2)}))$, $z^{(3)} = \mathcal{G}(\mathcal{F}_\Xi(x^{(3)}))$ and $z^{(4)} = \mathcal{G}(\mathcal{F}_\Xi(x^{(4)}))$.

Based on the above representations, we conduct multi-object correspondence contrast from three aspects. First, the contrast between multi-object image and single object one is implemented to model the representations of the corresponding objects in the image. The objective function is computed by cross entropy metric between similarity score targets and the predictions as

$$\mathcal{L}_{\mathrm{m2s}}(p^{\mathrm{mul1}}, z^{(3)}) = -\frac{1}{N \cdot r^2} \sum_{i=0}^{N-1} \sum_{j=0}^{r^2-1} \log \frac{\exp(\mathrm{sim}\,(p_i^{\mathrm{mul1}}, z_{y_{ij}^{\mathrm{m2s}}}^{(3)})/\tau)}{\sum_{k=0}^{N-1} \exp(\mathrm{sim}\,(p_i^{\mathrm{mul1}}, z_k^{(3)})/\tau)}, \tag{11}$$

where $\mathrm{sim}\,(h, z) = \frac{h^{\mathrm{T}} \cdot z}{\|h\| \cdot \|z\|}$ denotes cosine similarity between $h$ and $z$; $\tau$ denotes temperature coefficient.

Second, the contrast between multi-object images is introduced to capture the global representations of multi-object scenarios, which can be computed by cross entropy metric between similarity score targets and the predictions as

$$\mathcal{L}_{\mathrm{m2m}}(p^{\mathrm{mul1}}, z^{\mathrm{mul2}}) = -\frac{1}{N \cdot r^2} \sum_{i=0}^{N-1} \sum_{j=0}^{2r^2-2} \omega_{ij}^{\mathrm{m2m}} \cdot \log \frac{\exp(\mathrm{sim}\,(p_i^{\mathrm{mul1}}, z_{y_{ij}^{\mathrm{m2m}}}^{\mathrm{mul2}})/\tau)}{\sum_{k=0}^{N-1} \exp(\mathrm{sim}\,(p_i^{\mathrm{mul1}}, z_k^{\mathrm{mul2}})/\tau)}. \tag{12}$$

Third, we consider that there is the potential domain gap between the synthesized multi-object images and natural images. To alleviate the above domain gap, additional contrastive objective between

| Method | Backbone | Batch size | #Epoch | Finetune (%) | Linear (%) | kNN (%) |
|---|---|---|---|---|---|---|
| SimCLR (Chen et al., 2020a) | ViT-S/16 | 4096 | 300 | NA | 69.0 | NA |
| SwAV (Caron et al., 2020) | ViT-S/16 | 4096 | 300 | NA | 67.1 | NA |
| BYOL (Grill et al., 2020) | ViT-S/16 | 4096 | 300 | NA | 71.0 | NA |
| MoCo v3 (Chen et al., 2021) | ViT-S/16 | 4096 | 300 | 81.4 | 72.5 | 67.8 |
| DINO (Caron et al., 2021) | ViT-S/16 | 1024 | 300 | NA | 72.5 | 67.9 |
| DINO* (Caron et al., 2021) | ViT-S/16 | 1024 | 300 | NA | 76.1 | 72.8 |
| iBOT (Zhou et al., 2022) | ViT-S/16 | 1024 | 800 | NA | 76.2 | 72.4 |
| **MOS (ours)** | ViT-S/16 | 1024 | 300 | **82.8** | **77.6** | **73.5** |
| **MOS (ours)** | ViT-S/16 | 1024 | 800 | **83.5** | **77.9** | **74.2** |
| SimCLR (Chen et al., 2020a) | ViT-B/16 | 4096 | 300 | NA | 73.9 | NA |
| SwAV (Caron et al., 2020) | ViT-B/16 | 4096 | 300 | NA | 71.6 | NA |
| BYOL (Grill et al., 2020) | ViT-B/16 | 4096 | 300 | NA | 73.9 | NA |
| MoCo v3 (Chen et al., 2021) | ViT-B/16 | 4096 | 300 | 83.2 | 76.5 | 70.7 |
| DINO (Caron et al., 2021) | ViT-B/16 | 1024 | 400 | NA | 72.8 | 68.9 |
| DINO* (Caron et al., 2021) | ViT-B/16 | 1024 | 400 | 82.3 | 78.2 | 76.1 |
| iBOT (Zhou et al., 2022) | ViT-B/16 | 1024 | 400 | NA | 76.0 | 71.2 |
| **MOS (ours)** | ViT-B/16 | 1024 | 300 | **84.5** | **79.2** | **76.6** |

Table 1: Performance (%) comparison on ImageNet-1K dataset under finetune, linear and kNN evaluation protocols. "NA" denotes that the result is not available in original paper. "*" denotes the model pretrained with multi-crop strategy (Caron et al., 2020). For fairness, we compare our method with iBOT without multi-crop strategy.

natural single object images is introduced, which can be presented as

$$\mathcal{L}_{\text{s2s}}(p^{(3)}, z^{(4)}) = -\frac{1}{N} \sum_{i=0}^{N-1} \log \frac{\exp(\text{sim}\,(p_i^{(3)}, z_i^{(4)})/\tau)}{\sum_{k=0}^{N-1} \exp(\text{sim}\,(p_i^{(3)}, z_k^{(4)})/\tau)}. \tag{13}$$

Furthermore, to reduce the risk of representation degeneration, stop gradient operation $\text{sg}(\cdot)$ (Grill et al., 2020) is applied to all projection representations $z$. The total contrastive objective for multi-object representation learning can be written as

$$\mathcal{L}_{\text{total}} = \mathcal{L}_{\text{m2s}}(p^{\text{mul1}}, \text{sg}(z^{(3)})) + \mathcal{L}_{\text{m2m}}(p^{\text{mul1}}, \text{sg}(z^{\text{mul2}})) + \mathcal{L}_{\text{s2s}}(p^{(3)}, \text{sg}(z^{(4)})). \tag{14}$$

## 4 EXPERIMENTS

### 4.1 EXPERIMENTAL SETTINGS

To evaluate the effectiveness of our proposed method, we conduct experiments on large-scale dataset: ImageNet-1K (Russakovsky et al., 2015) and small-scale datasets: CIFAR10 and CIFAR100 (Krizhevsky et al., 2009). To further validate the effectiveness of our method on multi-object representations, we transfer the model pretrained on ImageNet-1K to object detection and instance segmentation tasks on COCO dataset (Lin et al., 2014), which contains 330K multi-object images. During pretraining, the images of CIFAR and ImageNet-1K are resized into $32 \times 32$ and $224 \times 224$, respectively. More details about data augmentation are depicted in the appendix.

We adopt basic ViT (Dosovitskiy et al., 2021) as backbone. The backbones for CIFAR contain ViT-T/2, ViT-S/2 and ViT-B/2, whose patch size for patchify is $2 \times 2$. The backbones for ImageNet-1K contain ViT-S/16 and ViT-B/16, whose patch size is $16 \times 16$. Following MoCo-v3 (Chen et al., 2021), additional MLP based projection and prediction module are adopted. More training details including hyper-parameters for ImageNet-1K, CIFAR and COCO, are depicted in the supplementary material. For a more comprehensive evaluation, we report all three evaluation protocols: finetune evaluation, linear evaluation and kNN (k nearest neighbor) evaluation. For both finetune and linear evaluation protocols, all pretrained models are finetuned for 100 epochs by default.

### 4.2 EXPERIMENTAL RESULTS

#### 4.2.1 IMAGE CLASSIFICATION ON IMAGENET-1K

To verify the effectiveness of our method, we evaluate the representation performance on single object centric images in ImageNet-1K. In Table 1, our method using ViT-S/16 respectively achieves

| Method | Backbone | #Epoch | Batch | CIFAR10 | | | CIFAR100 | | |
|---|---|---|---|---|---|---|---|---|---|
| | | | | Tune | Linear | kNN | Tune | Linear | kNN |
| MoCo v3 (Chen et al., 2021) | ViT-T/2 | 800 | 512 | 95.5 | 89.8 | 88.1 | 78.6 | 67.1 | 58.9 |
| DINO (Caron et al., 2021) | ViT-T/2 | 800 | 512 | 93.5 | 88.4 | 87.3 | 75.4 | 61.8 | 57.4 |
| iBOT (Zhou et al., 2022) | ViT-T/2 | 800 | 512 | 96.3 | 93.0 | 92.3 | 82.0 | 63.6 | 58.2 |
| **MOS (ours)** | ViT-T/2 | 800 | 512 | **97.6** | **94.8** | **93.2** | **83.7** | **73.5** | **67.8** |
| MoCo v3 (Chen et al., 2021) | ViT-S/2 | 800 | 512 | 95.8 | 90.2 | 89.1 | 78.8 | 66.4 | 62.3 |
| DINO (Caron et al., 2021) | ViT-S/2 | 800 | 512 | 96.3 | 92.6 | 91.8 | 75.9 | 63.8 | 60.6 |
| iBOT (Zhou et al., 2022) | ViT-S/2 | 800 | 512 | 97.0 | 94.8 | 93.1 | 82.8 | 67.8 | 64.2 |
| **MOS (ours)** | ViT-S/2 | 800 | 512 | **98.3** | **96.3** | **95.1** | **86.1** | **78.5** | **73.5** |
| MoCo v3 (Chen et al., 2021) | ViT-B/2 | 800 | 512 | 96.1 | 90.9 | 89.5 | 79.5 | 67.3 | 63.4 |
| DINO (Caron et al., 2021) | ViT-B/2 | 800 | 512 | 96.8 | 92.8 | 92.1 | 76.4 | 65.6 | 62.7 |
| iBOT (Zhou et al., 2022) | ViT-B/2 | 800 | 512 | 97.3 | 94.9 | 93.2 | 83.3 | 68.7 | 65.5 |
| **MOS (ours)** | ViT-B/2 | 800 | 512 | **98.3** | **96.4** | **95.1** | **86.2** | **79.6** | **74.4** |

Table 2: Performance (%) comparison on CIFAR datasets using finetune, linear and kNN evaluation protocols, respectively. "Tune" denotes classification accuracy using finetune protocol.

82.8%, 77.6% and 73.5% accuracy under finetune, linear and kNN protocol, when the model is pretrained for 300 epochs. Under the same pretraining epochs, our method surpasses the counterpart, DINO without multi-crop strategy by a large margin, 5.1% linear accuracy and 5.6% kNN accuracy. Furthermore, our method even outperforms DINO with multi-crop strategy when trained for 300 epochs. When trained using ViT-S/16 for 800 epochs, our method outperforms state-of-the art, iBOT by 1.7% and 1.8% under linear and finetune protocol, respectively. For ViT-B/16 backbone, our method trained for 300 epochs achieves significantly better performance other counterpart methods trained for 400 epochs, e.g., 3.2% linear accuracy and 5.4% kNN accuracy improvement over iBOT. Even though our method is trained to model multi-object representations, the experimental results illustrate that our method also presents excellent performance on single object centric classification tasks.

### 4.2.2 IMAGE CLASSIFICATION ON CIFAR

To investigate the generality of our method on small datasets, we further evaluate our method on CIFAR10 and CIFAR100, where all ViT models are pretrained on CIFAR datasets from scratch. As shown in Table 2, our method achieves significant performance improvement over other contrastive counterparts under all evaluation protocols on CIFAR10 and CIFAR100. In particular, our method with ViT-S/2 outperforms the previous self-supervised method, iBOT by a significantly large margin, 3.3% finetune accuracy, 10.7% linear accuracy and 9.3% kNN accuracy gain on CIFAR100. Despite extremely data-hungry property of ViT family architecture, our method consistently achieves remarkable improvement with the increment of model scaling from ViT-S/2 to ViT-B/2 on small dataset, CIFAR100. We believe that our proposed multiple object stitching strategy can effectively reduce the overfitting risk of ViT model on small datasets. Meanwhile, the experimental results further support that our method can consistently improve the representation performance on single object centric images.

### 4.2.3 OBJECT DETECTION AND INSTANCE SEGMENTATION

To validate the effectiveness of our method on multi-object image representation, we evaluate our method on object detection and instance segmentation tasks of COCO dataset (Lin et al., 2014), where most of the images contain multiple objects. For better fairness, two network architectures with similar number of parameters, ResNet50 and ViT-S/16 are adopted as th backbone of Mask RCNN (He et al., 2017) for comparison. As shown in Table 3, our method achieves the best results with 45.6 bounding box AP ($AP^{bb}$) and 40.6 mask AP ($AP^{mk}$) when the model is pretrained for 300 epochs. Our method significantly outperforms both contrastive methods for single object centric representation (MoCo v3 (Chen et al., 2021) and DINO (Caron et al., 2021)) and the ones for multi-object representation (SelfPatch (Yun et al., 2022) and ADCLR (Zhang et al., 2023)). The above experimental results support that our proposed multiple object stitching strategy can effectively capture the representations of multiple objects in image and improve the transfer learning performance on downstream dense prediction tasks, object detection and instance segmentation.

| Method | Backbone | #Epoch | #Param | Detection | | | Segmentation | | |
|---|---|---|---|---|---|---|---|---|---|
| | | | | $AP^{bb}$ | $AP^{bb}_{50}$ | $AP^{bb}_{75}$ | $AP^{mk}$ | $AP^{mk}_{50}$ | $AP^{mk}_{75}$ |
| MoCo v2 (Chen et al., 2020b) | ResNet50 | 200 | 25.6M | 39.2 | 58.8 | 42.5 | 34.3 | 55.5 | 36.6 |
| SwAV (Caron et al., 2020) | ResNet50 | 200 | 25.6M | 37.6 | 57.6 | 40.3 | 33.1 | 54.2 | 35.1 |
| BYOL (Grill et al., 2020) | ResNet50 | 200 | 25.6M | 37.9 | 57.8 | 40.9 | 33.2 | 54.3 | 35.0 |
| DenseCL (Wang et al., 2021) | ResNet50 | 200 | 25.6M | 39.6 | 59.3 | 43.3 | 35.7 | 56.5 | 39.2 |
| ReSim (Xiao et al., 2021) | ResNet50 | 200 | 25.6M | 39.5 | 59.9 | 43.3 | 35.8 | 57.0 | 38.4 |
| DetCo (Xie et al., 2021a) | ResNet50 | 200 | 25.6M | 40.1 | 61.0 | 43.9 | 36.4 | 58.0 | 38.9 |
| MoCo v3 (Chen et al., 2021) | ViT-S/16 | 300 | 22.0M | 39.8 | 62.6 | 43.1 | 37.1 | 59.6 | 39.2 |
| DINO (Caron et al., 2021) | ViT-S/16 | 300 | 22.0M | 40.8 | 63.4 | 44.2 | 37.3 | 59.9 | 39.5 |
| SelfPatch (Yun et al., 2022) | ViT-S/16 | 300 | 22.0M | 42.1 | 64.9 | 46.1 | 38.5 | 61.3 | 40.8 |
| ADCLR (Zhang et al., 2023) | ViT-S/16 | 300 | 22.0M | 44.3 | 65.4 | 47.6 | 39.7 | 62.1 | 41.5 |
| **MOS (ours)** | ViT-S/16 | 300 | 22.0M | **45.6** | **65.9** | **48.9** | **40.6** | **62.8** | **42.6** |

Table 3: Object detection and instance segmentation on COCO dataset. Mask RCNN is adopted.

## 4.3 ABLATION STUDY

### 4.3.1 THE EFFECT OF LOSS FUNCTION

To validate the effectiveness of each loss item in our method, we evaluate the linear accuracy and kNN accuracy of ViT-S model pretrained by different loss combinations as Table 4. For CIFAR100, the combination of multiple-to-single loss $\mathcal{L}_{m2s}$ and single-to-single loss $\mathcal{L}_{s2s}$ yields 9.1% linear accuracy and 7.8% kNN accuracy improvement over the one with only single-to-single loss $\mathcal{L}_{s2s}$. For CIFAR10 and ImageNet-1K, the introduce of multiple-to-single loss $\mathcal{L}_{m2s}$ also produces large improvement over the baseline with only single-to-single loss $\mathcal{L}_{s2s}$ or only multiple-to-single loss $\mathcal{L}_{m2s}$. It demonstrates that our proposed multiple-to-single loss can effectively improve the quality of unsupervised representation.

| $\mathcal{L}_{s2s}$ | $\mathcal{L}_{m2s}$ | $\mathcal{L}_{m2m}$ | CIFAR10 | | CIFAR100 | | ImageNet-1K | | COCO | |
|---|---|---|---|---|---|---|---|---|---|---|
| | | | Linear | kNN | Linear | kNN | Linear | kNN | $AP^{bb}$ | $AP^{mk}$ |
| ✓ | – | – | 90.1 | 89.2 | 65.7 | 62.8 | 73.2 | 68.3 | 42.8 | 38.9 |
| – | ✓ | – | 90.5 | 89.7 | 68.9 | 63.6 | 72.8 | 67.9 | 43.1 | 38.8 |
| – | – | ✓ | NA | NA | NA | NA | NA | NA | NA | NA |
| – | ✓ | ✓ | 92.7 | 90.3 | 71.2 | 68.4 | 74.2 | 73.5 | 43.6 | 39.2 |
| ✓ | ✓ | – | 94.6 | 93.3 | 74.8 | 70.6 | 76.2 | 72.1 | 44.3 | 39.7 |
| ✓ | ✓ | ✓ | **96.3** | **95.1** | **78.5** | **73.5** | **77.6** | **73.5** | **45.6** | **40.6** |

Table 4: The effect of loss items on CIFAR and ImageNet-1K dataset with ViT-S backbone. The performance (%) is measured by linear and kNN evaluation. "NA" denotes that the result is not available since the training of the model fails to converge during pretraining.

When the model is pretrained by only multiple-to-single loss $\mathcal{L}_{m2s}$, the improvement over the baseline with single-to-single loss $\mathcal{L}_{s2s}$ is limited. For ImageNet-1K, the pure multiple-to-single loss $\mathcal{L}_{m2s}$ is even worse than the one with single-to-single loss $\mathcal{L}_{s2s}$. Moreover, the model trained by only multiple-to-multiple loss $\mathcal{L}_{m2m}$ fails to converge. This can be in part explained by that, single-to-single loss $\mathcal{L}_{s2s}$ can reduce representation gap between the synthesized images and the natural one. When all three loss $\mathcal{L}_{s2s}$, $\mathcal{L}_{m2s}$ and $\mathcal{L}_{m2m}$ are applied, our method achieves the best results on all three datasets, CIFAR10, CIFAR100, ImageNet-1K and COCO, significantly surpassing the baseline with only $\mathcal{L}_{s2s}$. The experimental results illustrate that the fusion of the above three losses can significantly improve the unsupervised representations, including the representations of single object centric images.

### 4.3.2 THE EFFECT OF SCALE FACTOR

To analyze the effect of factor $r$ and $s$, we evaluate different sampling ranges for the pretraining of ViT-S/16 on ImageNet-1K and COCO dataset. As Table 5, compared to baseline with $r = 1$ and $s = 1$, sampling $r$ from $\{1, 2\}$ achieves 3.6% linear accuracy and 4.0% kNN accuracy improvement on ImageNet-1K, 2.9 $AP^{bb}$ and 1.7 $AP^{mk}$ improvement on COCO. It effectively supports that our proposed object stitching strategy can significantly improve the representation quality for both single object centric images and multi-object ones. When we sample $s$ from $\{1, 2\}$, the performance on both ImageNet-1K and COCO is further improved, especially 3.3 $AP^{bb}$ and 2.1 $AP^{mk}$ improvement on COCO. This result demonstrates that richer scale factors in synthesized multi-object image can further improve the representation performance and prove especially effective in the multi-object scenarios of COCO. Moreover, larger $r$ factor sampling range as $\{1, 2, 3\}$ doesn't introduce additional performance gain. In summary, the sampling of scale factor $r$ and $s$ effectively simulates

multi-scale multi-object scenarios in natural images and improves the unsupervised representation performance on both single object centric images and multi-object ones.

| $r$ factor sampling range | $s$ factor sampling range | ImageNet-1K | | COCO | |
|---|---|---|---|---|---|
| | | Linear | kNN | $AP^{bb}$ | $AP^{mk}$ |
| $\{1\}$ | $\{1\}$ | 73.2 | 68.3 | 39.4 | 36.8 |
| $\{1, 2\}$ | $\{1\}$ | 76.8 | 72.3 | 42.3 | 38.5 |
| $\{1, 2\}$ | $\{1, 2\}$ | **77.6** | **73.5** | **45.6** | **40.6** |
| $\{1, 2, 3\}$ | $\{1, 2\}$ | 76.9 | 72.6 | 42.1 | 38.2 |

Table 5: The effect of factor sampling range on ImageNet-1K and COCO dataset with ViT-S/16 backbone.

## 5 DISCUSSION

Our proposed method synthesizes multi-object image by stitching off-the-shelf single object centric images, inevitably introducing artificiality produced by the boundary between stitched objects. Therefore, there is potential domain gap between our synthesized multi-object images and the natural ones, which may affect the multi-object representation learning. To this end, we introduce the contrastive objective between natural single object centric images to alleviate the above issue, which effectively improves the multi-object representation performance in the experiments.

## 6 CONCLUSION

In this paper, we investigate two issues of unsupervised multi-object representation learning, namely semantics inconsistency between views and inaccurate object correspondence. To this end, we propose a novel unsupervised representation learning method to learn multi-object-aware representations by stitching off-the-shelf single object centric images into a synthesized multi-object one. Experimental results demonstrate that our method achieves state-of-the-art performance on various downstream tasks, including image classification, object detection and instance segmentation. Notably, compared to the previous unsupervised multi-object representation learning methods, our method keeps very competitive performance on image classification task, while achieving state-of-the-art results on object detection and instance segmentation. In future work, we plan to apply our method on more modalities, e.g. audio and video, to model more complicated scenario representations in an unsupervised manner.

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

## A APPENDIX

## B MULTIPLE OBJECT STITCHING

In this section, we give more explanations about multiple object stitching strategy. As shown in Figure 4 (a), we stitch the given image batch $\mathcal{V} = \{\{\mathcal{V}_{ij}\}_{j=0}^{r^2-1}\}_{i=0}^{N-1}$ by rearranging the elements by the indices of the first dimension $u(i,j) = (i+j) \bmod N$ as $\mathcal{I}_i = \text{stitch}\left(\{\mathcal{V}_{u(i,j)j}\}_{j=0}^{r^2-1}, r\right)$. When the index $(i+j)$ overflows the boundary along the first dimension, we introduce modular operation to index $(i+j)$ as $((i+j) \bmod N)$ to cyclically stitch the object views $\{\mathcal{V}_{u(i,j)j}\}_{j=0}^{r^2-1}$ in the image batch $\mathcal{V}$.

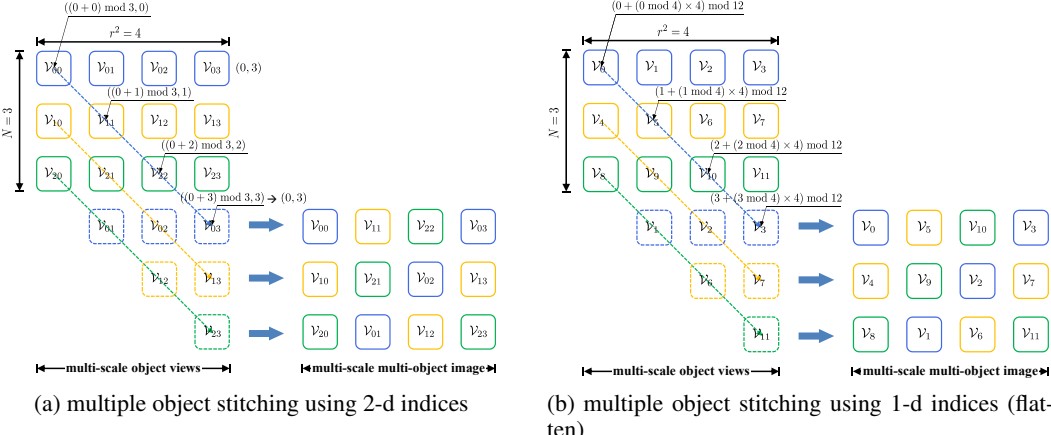

(a) multiple object stitching using 2-d indices

(b) multiple object stitching using 1-d indices (flatten)

Figure 4: The illustration for multiple object stitching strategy, where $r = 2$, $s = 1$ and the batch size $N = 3$.

To fit our multiple object stitching strategy into efficient implementation of tensor operation, we flatten the indices of object views $\mathcal{V} = \{\{\mathcal{V}_{ij}\}_{j=0}^{r^2-1}\}_{i=0}^{N-1}$ as $t = \{\{i \cdot r^2 + j\}_{j=0}^{r^2-1}\}_{i=0}^{N-1}$ in Figure 4 (b). To stitch the object views $\mathcal{V}$, the index transformation on the corresponding 1-d indices can be presented as

$$q = (t + (t \bmod r^2) \cdot r^2) \bmod T, \tag{15}$$

where $T = N \cdot r^2$ denotes the number of object views in $\mathcal{V}$. Finally, our multiple object stitching strategy can be effectively implemented by tensor operation, $\mathcal{I} = \text{index}(\mathcal{V}, q)$.

As shown in Figure 5, we give the computation of multiple-to-multiple targets and their corresponding similarity scores. Without loss of generality, we analyze the multiple-to-multiple targets for the second sample in view 1, including the object views from image sample $(1, 2, 3)$. The stitched samples in view 2, which contains the objects from image sample $(1, 2, 3)$, includes 5 samples, $(8, 0, 1)$, $(0, 1, 2)$, $(1, 2, 3)$, $(2, 3, 4)$ and $(3, 4, 5)$. Hence, the multiple-to-multiple target between view 1 and view 2 is $((0 - 2, 0 - 1, 0, 0 + 1, 0 + 2) + 9) \bmod 9 = (7, 8, 0, 1, 2)$. The general formula for multiple-to-multiple targets can be written as

$$y^{\text{m2m}} = \{\{l\}_{l=(i-r^2+1+N) \bmod N}^{(i+r^2-1) \bmod N}\}_{i=0}^{N-1}, \tag{16}$$

which contains $(2M - 1)$ positive samples for view 1. As shown in Figure 5, the number of overlapping objects between positive samples and the sample in view 1 are different, $(\frac{1}{3}, \frac{2}{3}, 1, \frac{2}{3}, \frac{1}{3})$ for image $(7, 8, 0, 1, 2)$, respectively. The general formula for multiple-to-multiple scores can be obtained by

$$\omega^{\text{m2m}} = \{\{1 - |r^2 - l - 1| / r^2\}_{l=0}^{2r^2-2}\}_{i=0}^{N-1}, \tag{17}$$

which provides more inter-scenario similarities for multiple object representation learning.

| multi-object view 1 | multi-object view 2 | multiple-to-multiple targets | similarity scores |
|---|---|---|---|
| | (8, 0, 1) | | |
| (0, 1, 2) | (0, 1, 2) | ((0-2, 0-1, 0, 0+1, 0+2) + 9) mod 9 | (1/3, 2/3, 1, 2/3, 1/3) |
| (1, 2, 3) | (1, 2, 3) | ((1-2, 1-1, 1, 1+1, 1+2) + 9) mod 9 | (1/3, 2/3, 1, 2/3, 1/3) |
| (2, 3, 4) | (2, 3, 4) | ((2-2, 2-1, 2, 2+1, 2+2) + 9) mod 9 | (1/3, 2/3, 1, 2/3, 1/3) |
| (3, 4, 5) | (3, 4, 5) | ((3-2, 3-1, 3, 3+1, 3+2) + 9) mod 9 | (1/3, 2/3, 1, 2/3, 1/3) |
| (4, 5, 6) | (4, 5, 6) | ((4-2, 4-1, 4, 4+1, 4+2) + 9) mod 9 | (1/3, 2/3, 1, 2/3, 1/3) |
| (5, 6, 7) | (5, 6, 7) | ((5-2, 5-1, 5, 5+1, 5+2) + 9) mod 9 | (1/3, 2/3, 1, 2/3, 1/3) |
| (6, 7, 8) | (6, 7, 8) | ((6-2, 6-1, 6, 6+1, 6+2) + 9) mod 9 | (1/3, 2/3, 1, 2/3, 1/3) |
| (7, 8, 0) | (7, 8, 0) | ((7-2, 7-1, 7, 7+1, 7+2) + 9) mod 9 | (1/3, 2/3, 1, 2/3, 1/3) |
| (8, 0, 1) | (8, 0, 1) | ((8-2, 8-1, 8, 8+1, 8+2) + 9) mod 9 | (1/3, 2/3, 1, 2/3, 1/3) |

Figure 5: The illustration of multiple-to-multiple targets and multiple-to-multiple similarity scores, where the $r^2 = 3$ and batch size $N = 9$.

---

**Algorithm 1** Multiple Object Stitching for Unsupervised Representation Learning

---

**Input:** Training dataset $\mathcal{X}$; ViT model $\mathcal{F}_*$; projection head $\mathcal{G}$; prediction head $\mathcal{H}$; momentum coefficient $\mu$.

**Output:** The parameters $\Theta$ of ViT model $\mathcal{F}_*$.

1: Initialize the parameters of $\mathcal{F}_*$, $\mathcal{G}$ and $\mathcal{H}$;
2: Initialize the momentum parameters $\Xi \leftarrow \Theta$;
3: **for** batch $x$ in dataset $\mathcal{X}$ **do**
4:     Augment image batch by $x^{(1)}, x^{(2)}, x^{(3)}, x^{(4)} = \mathcal{T}_1(x), \mathcal{T}_2(x), \mathcal{T}_3(x), \mathcal{T}_4(x)$;
5:     Obtain multi-scale multi-object views by $\mathcal{I}^{(1)}, y^{\text{m2s1}} = \text{STITCH}(x^{(1)})$;
6:     Obtain multi-scale multi-object views by $\mathcal{I}^{(2)}, y^{\text{m2s2}} = \text{STITCH}(x^{(2)})$;
7:     Obtain label $y^{\text{m2m}}$ and weight $\omega^{\text{m2m}}$ by Eq. 9 & 10;
8:     Obtained prediction rep. by $p^{\text{mul1}}, p^{(3)} = \mathcal{H}(\mathcal{G}(\mathcal{F}_\Theta(\mathcal{I}^{(1)}))), \mathcal{H}(\mathcal{G}(\mathcal{F}_\Theta(x^{(3)})))$;
9:     Obtained projection rep. by $z^{\text{mul2}}, z^{(3)}, z^{(4)} = \mathcal{G}(\mathcal{F}_\Xi(\mathcal{I}^{(2)})), \mathcal{G}(\mathcal{F}_\Xi(x^{(3)})), \mathcal{G}(\mathcal{F}_\Xi(x^{(4)}))$;
10:     Compute loss $\mathcal{L}_{\text{total}}$ and update the parameters $\Theta$;
11:     Update momentum parameters by $\Xi \leftarrow \mu \cdot \Xi + (1 - \mu) \cdot \Theta$;
12: **end for**
13:
14: **function** STITCH(Batch $x$)                                  ▷ stitch operation
15:     **for** image $x_i$ in image batch $x$ **do**
16:         Augment $x_i$ into views with different scale factors $s$ by $\{\hat{x}_{ij}^{(s)}\}_{j=0}^{(s \cdot r)^2 - 1} = \mathcal{T}(x_i, r, s)$;
17:         Divide $\{\hat{x}_{ij}^{(s)}\}_{j=0}^{(s \cdot r)^2 - 1}$ into $r^2$ groups $\{G_{ij}^{(s)}\}_{j=0}^{r^2 - 1}$;
18:         Stitch the views in $G_{ij}^{(s)}$ into a scaled image by $\hat{G}_{ij}^{(s)} = \text{stitch}(G_{ij}^{(s)}, s)$;
19:         Randomly sample scale factor $s \in \{1, 2, \cdots, S\}$ to construct multi-scale view $\mathcal{V}_{ij}$;
20:     **end for**
21:     Flatten multi-scale views $\{\{\mathcal{V}_{ij}\}_{j=0}^{r^2-1}\}_{i=0}^{N-1}$ as $\mathcal{V} = \{\mathcal{V}_t\}_{t=0}^{T-1}$;
22:     Obtain the indices for object stitching $q$ by Eq. 7;
23:     Implement multi-object stitching by $\mathcal{I} = \text{index}(\mathcal{V}, q)$;
24:     Compute mixed targets $y^{\text{m2s}}$ by Eq. 8;
25:     **return** $\mathcal{I}, y^{\text{m2s}}$
26: **end function**

---

## C ALGORITHM

For better illustration, we present our multiple object stitching strategy as Algorithm 1.

## D IMPLEMENTATION DETAILS

All models are pretrained on 6 GPU servers, each with 8 Nvidia GeForce RTX 3090 GPUs. For ViT-S/16 on ImageNet-1K, the model is pretrained with AdamW optimizer. The learning rate is set

to $2 \times 10^{-3}$ with a batch size of 1024 and follows a cosine schedule with a warmup of 10 epochs. The weight decay also follows a cosine schedule from 0.04 to 0.4. For further regularization, we adopt DropConnect with drop rate of 0.1. The momentum coefficient for momentum encoder follows cosine schedule from 0.992 to 1.0. The projection and prediction module respectively contain 3-layer and 2-layer multi-layer perceptron (MLP) with hidden dimension of 4096 and output dimension of 256. The temperature coefficient is set to 0.2. For multiple object stitching, both $r$ and $s$ are randomly sampled from $\{1, 2\}$. The hyper-parameters for data augmentation are summarized as Table 6. For ViT-B/16 on ImageNet-1K, learning rate is set to $3 \times 10^{-3}$ with a batch size of 1024. The momentum coefficient for momentum encoder follows cosine schedule from 0.99 to 1.0. The drop rate of DropConnect is set to 0.3. Other hyper-parameters are consistent with the ones as ViT-S/16 on ImageNet-1K.

| Augmentation | Parameters | ImageNet-1K | | | |
|---|---|---|---|---|---|
| | | Aug. $\mathcal{T}_1(\cdot)$ | Aug. $\mathcal{T}_2(\cdot)$ | Aug. $\mathcal{T}_3(\cdot)$ | Aug. $\mathcal{T}_4(\cdot)$ |
| random crop and resize | area of the crop | [0.2, 1.0] | [0.2, 1.0] | [0.1, 1.0] | [0.1, 1.0] |
| | aspect ratio of the crop | $[\frac{3}{4}, \frac{4}{3}]$ | $[\frac{3}{4}, \frac{4}{3}]$ | $[\frac{3}{4}, \frac{4}{3}]$ | $[\frac{3}{4}, \frac{4}{3}]$ |
| | image size | 112 | 112 | 224 | 224 |
| | number of views | 4 | 4 | 1 | 1 |
| random color jittering | color jittering probability | 0.8 | 0.8 | 0.8 | 0.8 |
| | max brightness adjustment intensity | 0.4 | 0.4 | 0.4 | 0.4 |
| | max contrast adjustment intensity | 0.4 | 0.4 | 0.4 | 0.4 |
| | max saturation adjustment intensity | 0.2 | 0.2 | 0.2 | 0.2 |
| | max hue adjustment intensity | 0.1 | 0.1 | 0.1 | 0.1 |
| random gray scale | color dropping probability | 0.2 | 0.2 | 0.2 | 0.2 |
| random Gaussian blurring | Gaussian blurring probability | 0.1 | 1.0 | 0.1 | 1.0 |
| | sigma of Gaussian blurring | [0.1, 2.0] | [0.1, 2.0] | [0.1, 2.0] | [0.1, 2.0] |
| random solarization | solarization probability | 0.0 | 0.2 | 0.0 | 0.2 |
| random horizontal flip | horizontal flip probability | 0.5 | 0.5 | 0.5 | 0.5 |

Table 6: The parameters of data augmentations applied during self-supervised training. "[·, ·]" denotes the range for uniform sampling.

For CIFAR10 and CIFAR100, all cropped image sizes are $32 \times 32$. For ViT-T/2 on CIFAR10 and CIFAR100, the learning rate is set to $4 \times 10^{-3}$ with a batch size of 512 and follows a cosine schedule with a warmup of 100 epochs. The weight decay is 0.1. The momentum coefficient for momentum encoder follows cosine schedule from 0 to 1.0. Other hyper-parameters are consistent with the ones as ViT-S/16 on ImageNet-1K. For ViT-S/2 on CIFAR10 and CIFAR100, we set the learning rate to $1 \times 10^{-3}$ with a batch size of 512 and follows a cosine schedule with a warmup of 100 epochs. Other hyper-parameters are consistent with the ones as ViT-S/16 on ImageNet-1K. For ViT-B/2 on CIFAR10 and CIFAR100, learning rate is set to $1.5 \times 10^{-3}$ with a batch size of 512. Other hyper-parameters are consistent with the ones as ViT-B/16 on ImageNet-1K.

## E  PRETRAINING CURVES

To further understand our method, we present the pretraining curves of our total loss and all loss items in Figure 6 and Figure 7. For CIFAR100 dataset in Figure 6, all losses achieves stable convergence during pretraining. Notably, the value of single-to-single loss is larger than the ones of multiple-to-single and multiple-to-multiple losses. I analyze the above results as follows. Due to the supervision of multiple-to-single and multiple-to-multiple losses, the representations extracted by out method are encouraged to capture the similarities between different image instances, e.g., potential positive samples. Therefore, single-to-single loss cannot simply reduce the similarities of other potential positive pairs (not the given positive pairs) for lower loss value.

For more complicate ImageNet-1K dataset in Figure 7, the values of losses don't consistently decrease with the increment of pretraining epochs. However, we find that the performance of its representation keeps sustained improvement during the pretraining. We plan to further explore the insight behind this result in future work.

## F  THE VISUALIZATION OF PRETRAINED REPRESENTATIONS

To further assess the representation quality of our method, we visualize the representations of our method and iBOT using t-SNE. For better visualization, we select the representations of simple CI-

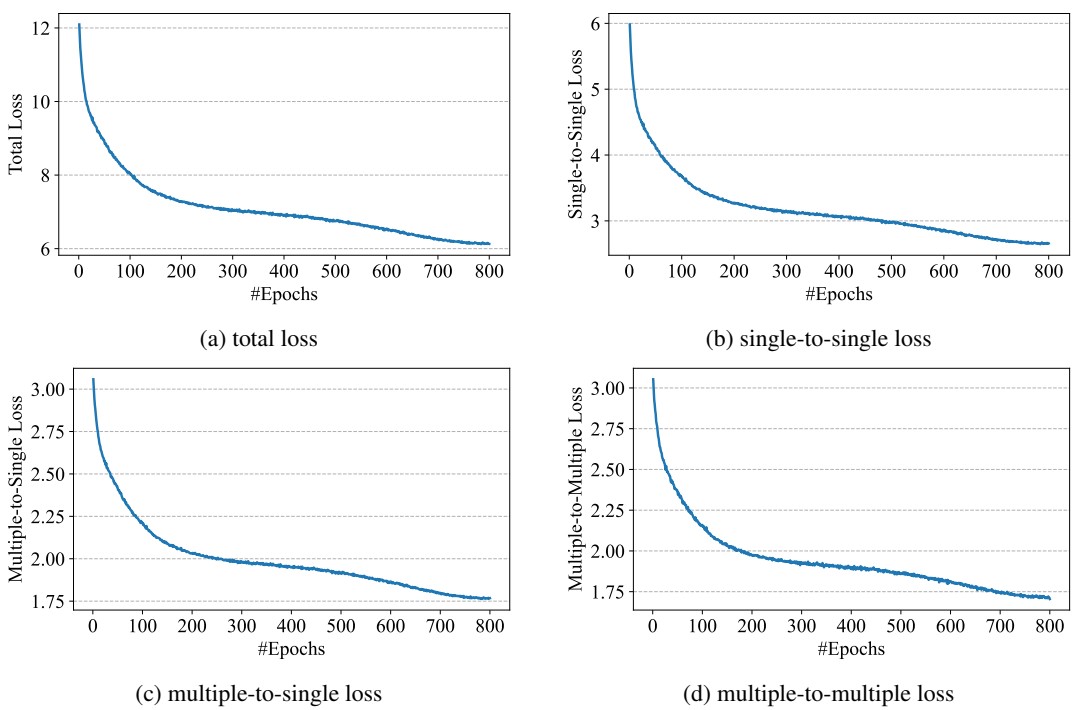

Figure 6: The pretraining curves for CIFAR100.

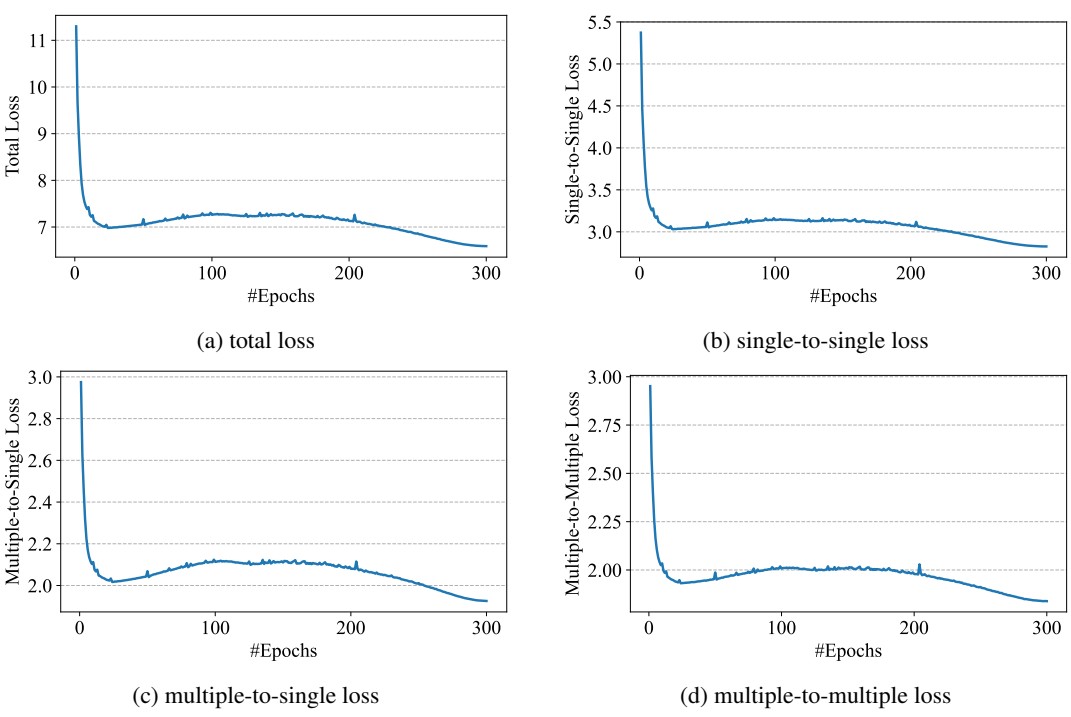

Figure 7: The pretraining curves for ImageNet-1K.

FAR10, which only contains 10 categories. For more complicated ImageNet-1K and CIFAR100 datasets, it's difficult to discriminate massive categories for better visualization by limited distinguished colored markers. As shown in Figure 8, our method can well group the samples from the

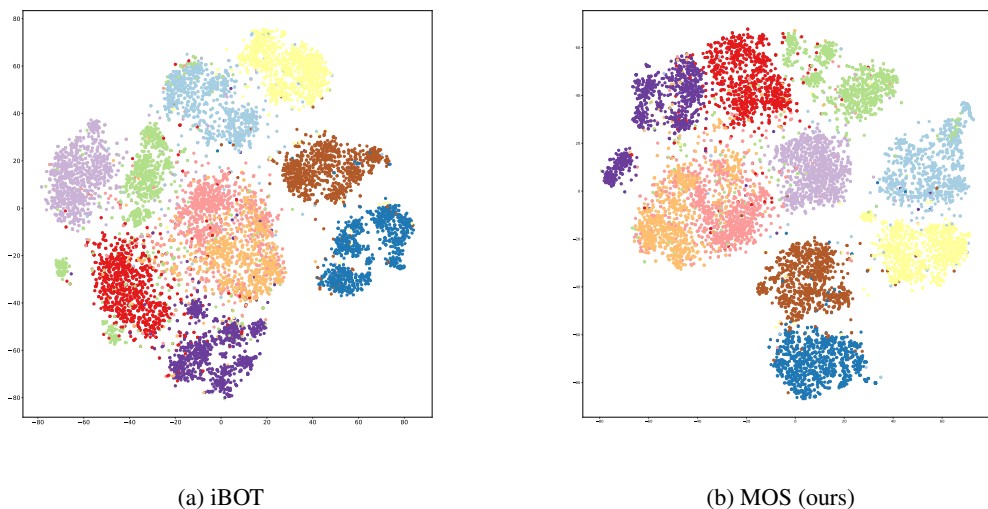

(a) iBOT          (b) MOS (ours)

Figure 8: The visualization of pretrained representations on CIFAR10 by t-SNE.

same categories in the representation space, in spite of without supervised training. Compared to iBOT, our method achieves better representation consistency for "green" category.

## G  THE VISUALIZATION OF SELF-ATTENTION MAP

To analyze the representations of our proposed method, we visualize the self-attention map of [CLS] token from multiple heads of the last transformer block (ViT-S/16 backbone on ImageNet-1K and COCO). The visualization results are shown in Figure 9, Figure 10 on ImageNet-1K, and Figure 11 on COCO, where each sample contains attention maps from 12 heads. Compared to the previous state-of-the-art, iBOT, the self-attention maps produced by our method present more clear semantics. Specifically, in the first row in Figure 10, the attention scores of our proposed MOS clearly focus on the fish objects, yet the attention scores obtained by iBOT are not so sharp on the position of the fishes. Moreover, in the third row in Figure 10, our proposed method successfully recognizes the reflection of shark, but the attentions produced by iBOT on the corresponding reflection are week.

## H  OBJECT CORRESPONDENCE

Following iBOT, we also visualize the correspondence between images with overlapping categories on COCO using our ViT-S/16 pretrained for 800 epochs in Figure 12.

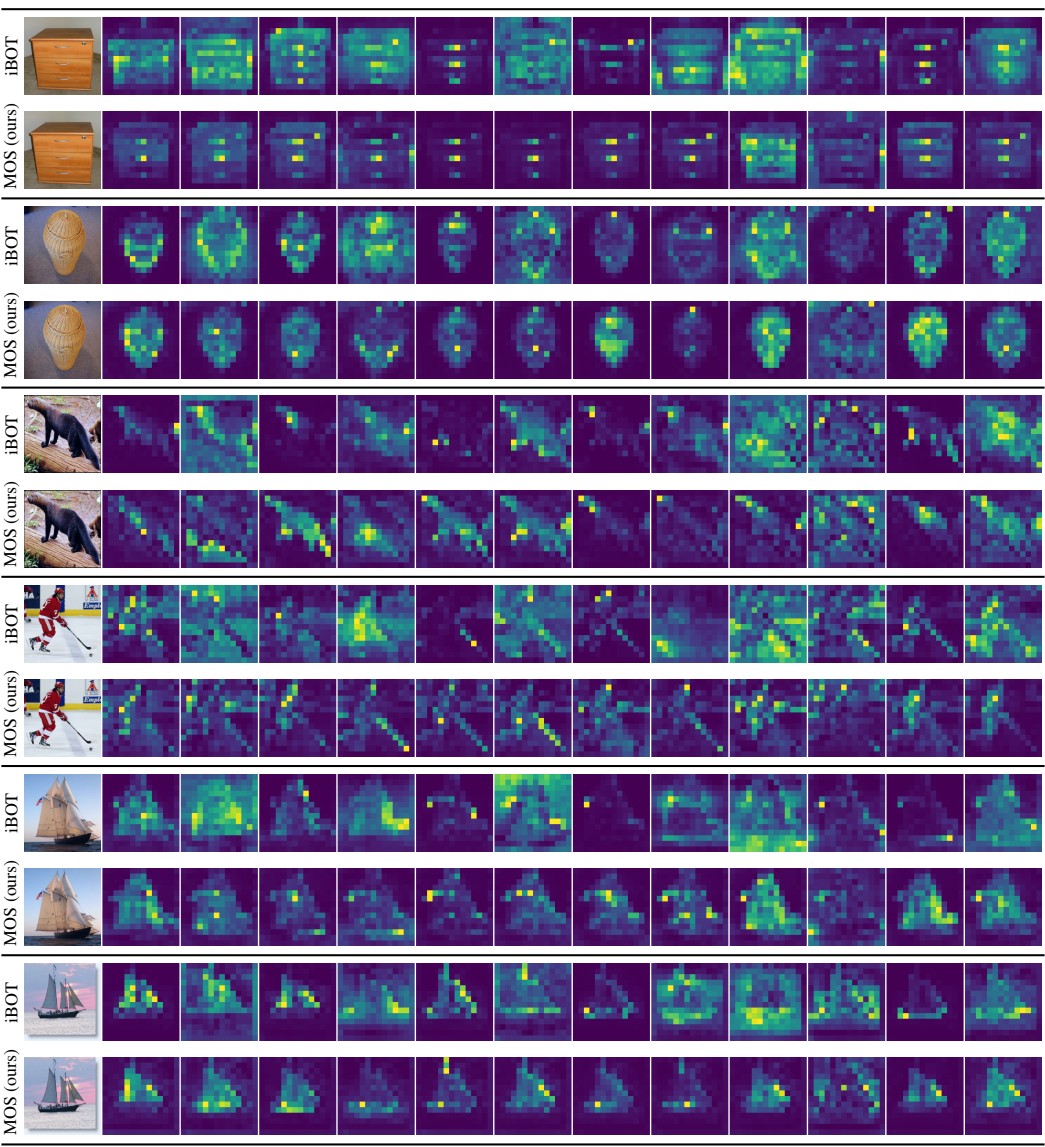

Figure 9: The visualization of self-attention maps on ImageNet-1K.

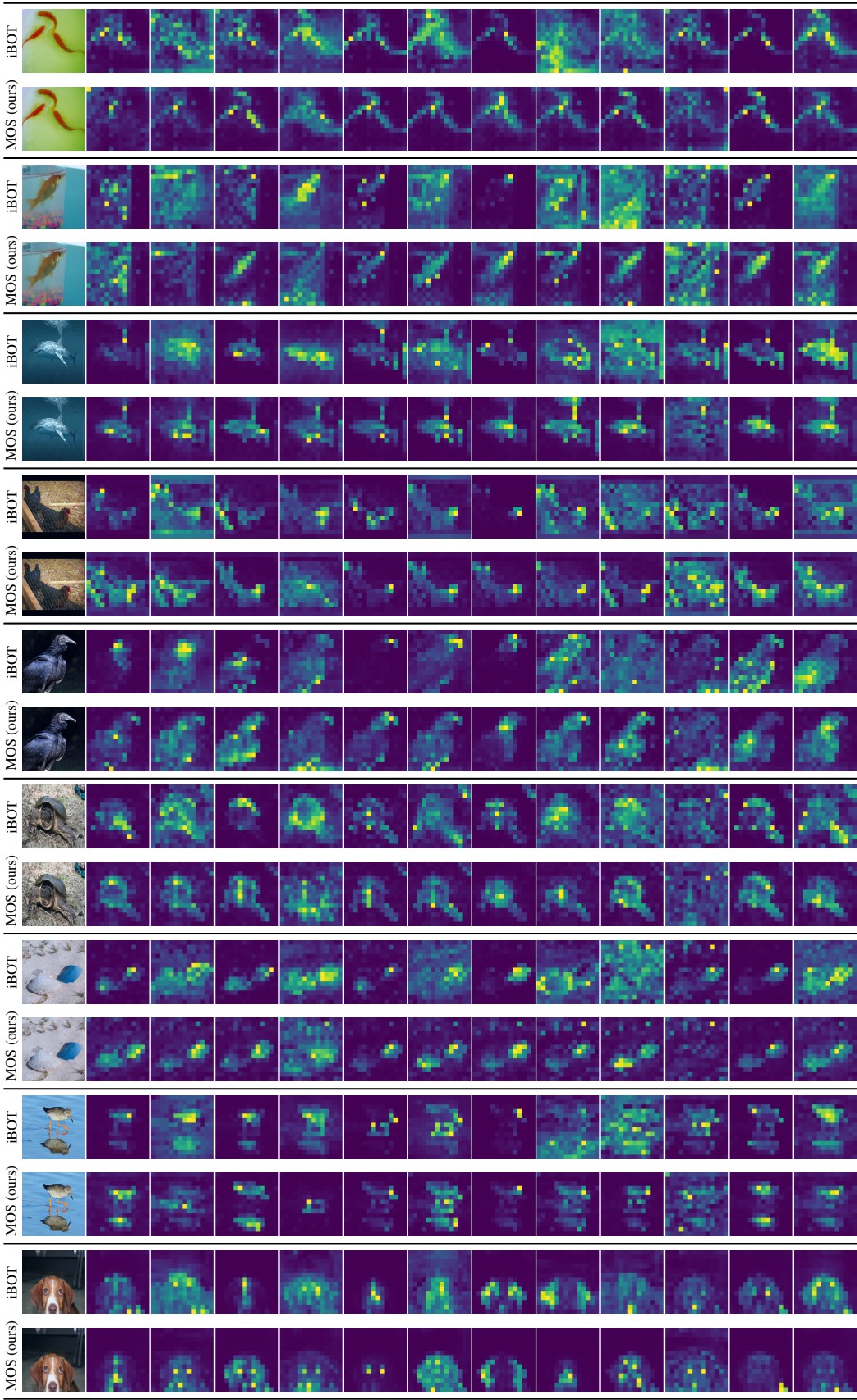

Figure 10: The visualization of self-attention maps on ImageNet-1K.

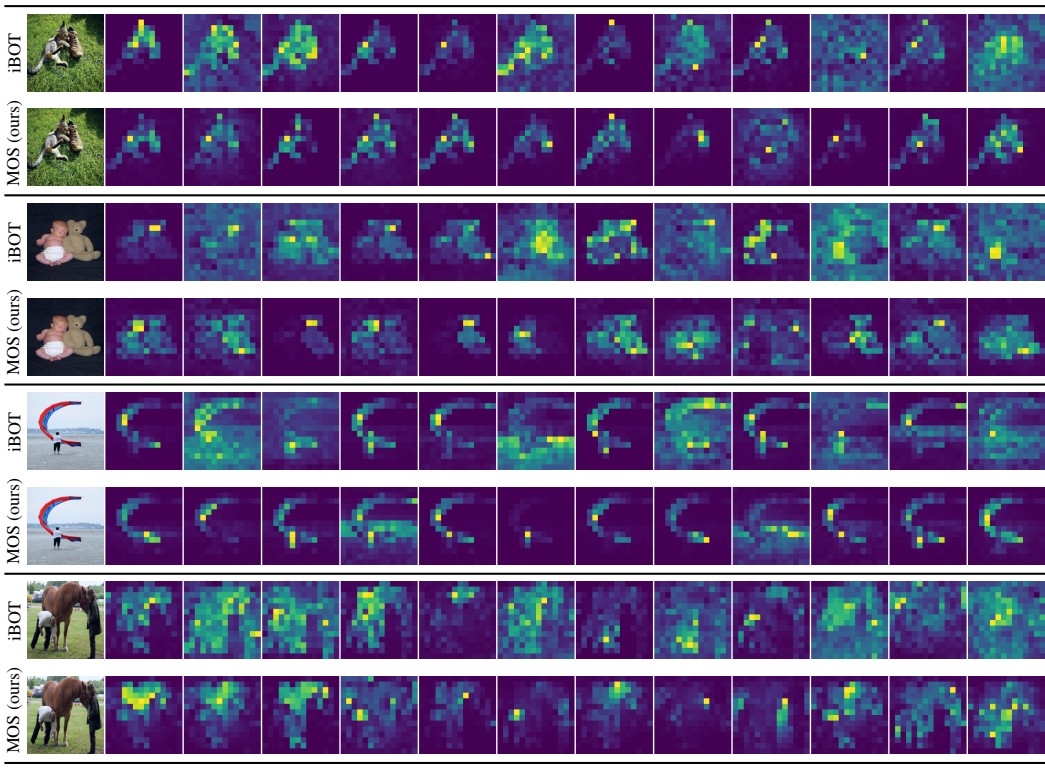

Figure 11: The visualization of attention maps on COCO using ViT-S/16 pretrained for 800 epochs.

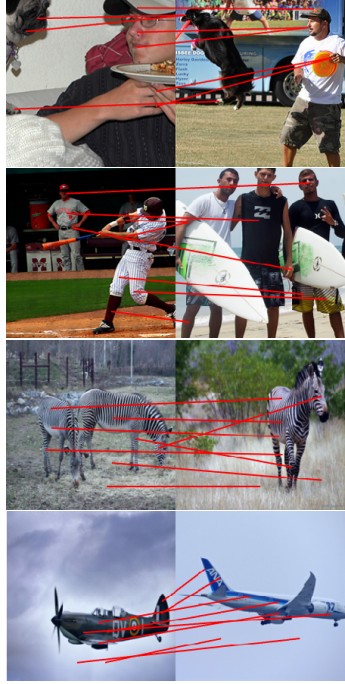
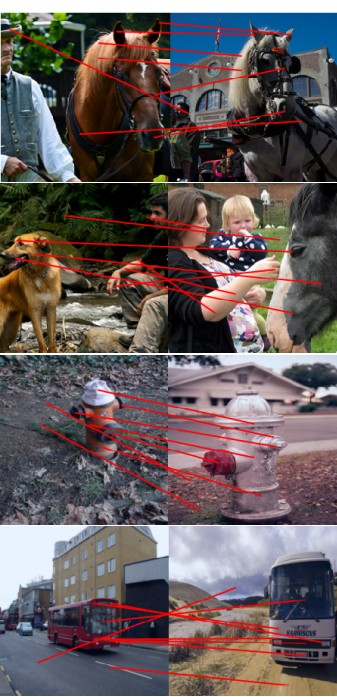

Figure 12: The correspondence visualization on COCO using ViT-S/16 pretrained for 800 epochs.

