# OpenReview forum: "Multiple Object Stitching for Unsupervised Representation Learning"
_ICLR.cc/2024/Conference — Submitted to ICLR 2024_

### Official Review · Reviewer_gfC6 · 2023-10-17

**Soundness:** 2 fair
**Presentation:** 2 fair
**Contribution:** 1 poor
**Rating:** 3
**Confidence:** 5

**Summary:**

This paper indicates that contrastive learning for single object centric images has achieved impressive performance and suffered inferior performance on multiple objects. To this end, this paper proposes a method, i.e., Multiple Object Stitching (MOS), to refine the unsupervised representation for multi-object images. Particularly, they construct the multi-object images by stitching the single object centric ones, where the objects in the synthesized multi-object images are predetermined. In the experiments, the proposed method is evaluated on multiple datasets.

**Strengths:**

Applying contrastive learning to multiple object scenarios is meaningful.

**Weaknesses:**

1. To the best of my knowledge, applying contrastive learning to multiple object scenarios has been explored by many works. There exist multiple works that aim to utilize contrastive loss for object detection and semantic segmentation. However, in the Introduction Section, the authors do not introduce these works. I recommend the authors modify their paper carefully, introduce these works, and make corresponding contrasts.

2. To address the multiple object contrastive learning, the authors propose a stitching strategy. However, this strategy is somewhat simple. Meanwhile, I am not clear on how to use the proposed strategy to obtain object-related content.

3. In Method Section, the authors should add a discussion section to discuss the advantages and disadvantages of the proposed method.  Using more stitched images may increase computational costs. The authors should give more interpretations. Finally, the authors should show some training curves, which is helpful for further understanding the proposed method. Meanwhile, the authors should show some T-SNE results to indicate the superiorities of the proposed method.

**Questions:**

The authors should evaluate the proposed method on object detection and semantic segmentation.

---

> ### Author Response · Authors · 2023-11-15
>
> Thanks for your constructive comments.  We address the concerns as follows:
>
> + **More introduction of related work**
>
>   Thanks for your constructive comments. Following the suggestions, we supplement more related works in the Introduction section.
>
> + **How to use the proposed strategy to obtain object-related content?**
>
>   The object-related images can be easily obtained from the off-the-shelf classification datasets, such as CIFAR10, CIFAR100 and ImageNet-1K. Due to single category label of each image from these datasets, the image context of these datasets can be regarded as single-object one.
>
> + **Discussion about advantages and disadvantages**
>
>   We supplement an additional section in the revision to discuss the advantages and disadvantages of our proposed method. The clarifications are presented as follows.
>
>   > Our proposed method synthesizes multi-object image by stitching off-the-shelf single object centric images, inevitably introducing artificiality produced by the boundary between stitched objects. Therefore, there is potential domain gap between our synthesized multi-object images and the natural ones, which may affect the multi-object representation learning. To this end, we introduce the contrastive objective between natural single object centric images to alleviate the above issue, which effectively improves the multi-object representation performance in the experiments.
>
> + **The increased computational cost of our method**
>
>   Actually, stitching more images doesn't introduce additional computational cost. Specifically, all stitched images are resized into 224×224 size, the same image size as the one adopted by plain contrastive learning methods. Hence, compared to plain image view for contrastive learning, our stitched images doesn't increase the cost of computation.
>
> + **Training curves of our method**
>
>   We supplement the pretraining curves in the appendix of our revised paper (**E Pretraining Curves** section). The clarifications are presented as follows.
>
>   > To further understand our method, we present the pretraining curves of our total loss and all loss items in Figure 6 and Figure 7. For CIFAR100 dataset in Figure 6, all losses achieves stable convergence during pretraining. Notably, the value of single-to-single loss is larger than the ones of multiple-to-single and multiple-to-multiple losses. I analyze the above results as follows.  Due to the supervision of multiple-to-single and multiple-to-multiple losses, the representations extracted by out method are encouraged to capture the similarities between different image instances, e.g., potential positive samples.  Therefore, single-to-single loss cannot simply reduce the similarities of other potential positive pairs (not the given positive pairs) for lower loss value.
>   >
>   > For more complicate ImageNet-1K dataset in Figure 7, the values of losses don't consistently decrease with the increment of pretraining epochs. However, we find that the performance of its representation keeps sustained improvement during the pretraining.   We plan to further explore the insight behind this result in future work.
>
> + **T-SNE visualization results**
>
>   We supplement the T-SNE visualization results in the appendix of our revised paper (**F The Visualization of Pretrained Representations** section). The clarifications are presented as follows.
>
>   > To further assess the representation quality of our method, we visualize the representations of our method and iBOT using t-SNE. For better visualization, we select the representations of simple CIFAR10, which only contains 10 categories. For more complicated ImageNet-1K and CIFAR100 datasets, it's difficult to discriminate massive categories for better visualization by limited distinguished colored markers. As shown in Figure 8, our method can well group the samples from the same categories in the representation space, in spite of without supervised training. Compared to iBOT, our method achieves better representation consistency for "green" category.
>
>
>
> + **The evaluation results on object detection and semantic segmentation**
>
>   Indeed, we have evaluated our pretrained model on object detection and instance segmentation tasks of COCO dataset in Table 3.
>
>   To further validate the effectiveness of our method, we conduct transfer learning experiments with the pretrained ViT-S/16 backbone using FPN architecture on semantic segmentation of ADE20K dataset. All models are finetuned for 40k iterations. The results are listed as follows.
>
>   |     Method     |   mIoU   |   aAcc   |   mAcc   |
>   | :------------: | :------: | :------: | :------: |
>   |   SelfPatch    |   41.2   |   80.7   |   52.1   |
>   |     ADCLR      |   42.4   |   81.1   |   54.2   |
>   | **MOS (ours)** | **43.9** | **82.4** | **55.8** |
>
>   The experimental results demonstrate that our method significantly outperform the previous method.

---

### Official Review · Reviewer_AZJE · 2023-10-26

**Soundness:** 2 fair
**Presentation:** 1 poor
**Contribution:** 2 fair
**Rating:** 5
**Confidence:** 4

**Summary:**

Contrastive learning suffers from the issue of semantic inconsistency caused by random cropping. This paper circumvents this problem by exclusively utilizing single-object images and their synthetic combinations, creating a 2x2 grid of a single image. This synthetic image serves as a pseudo-multi-object image, where the location of the positive object is known, allowing it to be trained similarly to a multi-object image. This technique enhances the learned representation, as evaluated in image classification and object detection benchmarks.

**Strengths:**

- Self-supervised learning, particularly learning from multi-object images, is an important problem.
- The performance improvement is quite significant, both in image classification and object detection.

**Weaknesses:**

**Movitation is not new**

The issue of semantic inconsistency in contrastive learning has been discussed in several prior works [1-4].
These works are not properly cited, which may lead readers to overestimate the contribution of this paper.
The efforts of prior work and the contributions of this paper should be clarified in the second paragraph of the introduction.

[1] CASTing Your Model: Learning to Localize Improves Self-Supervised Representations. CVPR'21.\
[2] Unsupervised Object-Level Representation Learning from Scene Images. NeurIPS'21.\
[3] Object-aware Contrastive Learning for Debiased Scene Representation. NeurIPS'21. \
[4] Object-aware Cropping for Self-Supervised Learning. TMLR'22.


---
**Not addressing the raised issue**

This paper does not address the issue that has been raised: the problem of random cropping in multi-object images.
Instead, the paper solely utilizes single-object images and their combinations, while ignoring multi-object images.

This raises two problems:
- The paper can be considered a general augmentation strategy for contrastive learning, rather than specialized for multi-object images. This necessitates a proper revision of the storyline, as well as the baselines to compare with.
- Restricting the training images to single-object images significantly limits the applicability of the method to the majority of unlabeled images containing multiple objects. Thus, the proposed method should be compared with vanilla contrastive learning trained on a larger superset of images that also includes multi-object images.


---
**Method is not new**

Combining multiple images for data augmentation has been studied in multiple prior works [5-6].
Furthermore, similar augmentation strategies have also been applied in the context of contrastive learning [7-8].
The paper should properly cite these prior works and make comparisons with them, particularly the most relevant MosRep [8].

[5] RICAP: Random Image Cropping and Patching Data Augmentation for Deep CNNs. ACML'18.\
[6] CutMix: Regularization Strategy to Train Strong Classifiers with Localizable Features. ICCV'19.\
[7] i-Mix: A Domain-Agnostic Strategy for Contrastive Representation Learning. ICLR'21.\
[8] Mosaic Representation Learning for Self-supervised Visual Pre-training. ICLR'23.


---
**Method section could be revised**

The current method section is difficult to understand.
Since the overall method is relatively simple, the authors should consider a more effective way to convey the message:
- Super/subscripts are overused. Consider simplifying the notations.
- Equations are overused. Consider moving less critical ones (e.g., straightforward definitions like Eq. (1)) into the inline text and using the equation mode exclusively for important formulas (e.g., the definitions of m2s/m2m/s2s losses) to emphasize them.
- For Fig. 2, consider making the caption self-contained, possibly using natural language without math notations that are defined in the main text below.

**Questions:**

1. Compare this method with prior works such as i-Mix and MosRep.
2. This method can also be applied to multi-object images?

---

> ### Author Response · Authors · 2023-11-15
>
> Thanks for your constructive feedback. We handle your main concerns as follows:
>
> + **More introduction about related works**
>
>   Thanks for your constructive comments. Following the suggestions, we supplement more related works in the Introduction section and further refine the writing.
>
> + **The addressing of the issue about random cropping in multi-object images**
>
>   There are some misunderstandings of our proposed method.
>
>   Our proposed method is designed to avoid random cropping issue suffered by directly conducting contrastive learning methods on natural multi-object images. To this end, we construct synthetic multi-object images by stitching the existing single-object-centric images, where the context of synthetic multi-object images can be predetermined. Therefore, our method can well address the random cropping issue on multi-object images.
>
> + **The comparison with MosRep**
>
>   Thanks for pointing out the missing references. We will carefully analyze and compare MosRep in the final revision.
>
>   Compared to MosRep, the main contributions (differences) of our method can be summarized as follows.
>
>   1. Our method provides an effective contrastive learning solution for multi-object images and achieves very competitive performance on both single-object-centric and multi-object tasks. In contrast, MosRep is only designed for the task on single-object-centric tasks.
>   2. Our loss function is significantly different from MosRep. Our proposed contrastive objective models the accurate correspondences of multiple-to-single, multiple-to-multiple and single-to-single relations. The stitched images are addressed as a whole in our method. However, MosRep handles the mosaic representations as single objects by ROI Align.
>
>   Moreover, we compare our method with MosRep according to the official implementation **[R1]** (The official source code of [R1] fails to achieve the results reported in the paper). The results are listed as follows.
>
>   |     Method      |   #Views    | Backbone  |  Linear   |    kNN    |
>   | :-------------: | :---------: | :-------: | :-------: | :-------: |
>   | MosRep **[R1]** | 2×224+4×112 | ResNet-50 |   72.3%   |   61.3%   |
>   | **MOS (Ours)**  | 2×224+4×112 | ResNet-50 | **75.6%** | **70.6%** |
>
>   The above results demonstrate that our method consistently outperforms MosRep by a large margin on both linear probing and kNN probing protocols.
>
>   **Moreover, we supplement the analysis of MosRep and cite it in the Related Work section.**
>
>   **[R1]** https://github.com/DerrickWang005/MosRep
>
> + **The refinement of the writing about method section**
>
>   Thanks for your constructive suggestion.  We will refine these notations in the final version.
>
>   For the notations in the paper, we mainly focus on the accurate expression of our method. We confirm that there is no useless notation or equation in our paper. In spite of somewhat complex notations, these notations can help authors to better understand the technique details of our method without confusion. Of course, we will further refine the writing of method section in the final revision.
>
> + **The application on multi-object images**
>
>   As clarified in the paper, our method is designed to simulate multi-object images by single-object-centric ones. Hence, the direct application of our method on multi-object images has not been considered in this paper. We would like to research this topic in future work.

---

> ### Comment · Reviewer_AZJE · 2023-11-20
> **Response to the Rebuttal**
>
> Thank you for the response. I have carefully read the rebuttal and revision. However, my original concerns have not been addressed.
>
> - **More introduction about related works**\
> The main idea of this paper is to restrict the positive pairs of contrastive learning to avoid semantic inconsistency. This idea is highly relevant to the suggested [1-4], which still have not been discussed in the paper. These approaches belong to a different category than the region-level and pixel-level contrastive learning.
>
> - **The addressing of the issue about random cropping in multi-object images**\
> In essence, the paper avoids using multi-object images to sidestep the issue of semantic inconsistency. This approach largely overlooks the potential information that could be learned from such images, and bypasses the problem rather than addressing it. Given that the majority of real-world images contain multiple objects, how can one collect single-object images in a scalable manner?
>
> - **The comparison with MosRep**\
> Thank you for the additional experiments. The rebuttal claims that "Our method provides an effective contrastive learning solution for multi-object images and achieves very competitive performance on *both single-object-centric and multi-object* tasks. In contrast, MosRep is only designed for the task on single-object-centric tasks." However, I cannot find the experiments supporting this claim. Can the authors elaborate and demonstrate which parts of MOS make it effective for multi-object tasks, while MosRep cannot?
>
> - **The refinement of the writing about method section**\
> I did not claim that there are unnecessary notations or equations. However, providing excessive details, including those that are less essential, can distract the readers. For example, overloading equations with numerous superscripts and subscripts hampers readability. As one example, the paper uses subscripts $_i$ to indicate the $i$-th sample of the batch, but this $i$ is unnecessary when describing other parts; one can simply denote the sample as $x$ instead of $x_i$. The readers can still understand that the same approach can be extended to other samples in the batch. Delivering the core message in a readable form while maintaining precision is the art of mathematical writing.

---

> > ### Author Response · Authors · 2023-11-21
> >
> > Thanks for your feedback. We further address your main concerns as follows:
> >
> > + **More introduction about related works**
> >
> >   Following the suggestions, we have supplemented the introduction of the suggested [1-4] in the revision.
> >
> >
> >
> > + **The addressing of the issue about random cropping in multi-object images**
> >
> >   Thanks for your comments.
> >
> >   We approve that our method relies on the single-object-centric images and cannot be directly applicable to multi-object images.
> >
> >   Notably, our method indeed avoids random cropping issue of the previous solutions. It's the advantage of our method, instead of a defect.
> >
> >   Following the suggestion, we are happy to explore the random cropping issue of natural multi-object images in future work.
> >
> >
> >
> > + **The comparison with MosRep**
> >
> >   We are sorry for the inaccurate clarification "MosRep is only designed for the task on single-object-centric tasks". We will refine the comparison with MosRep in the final revision.
> >
> >   Our method indeed achieves more competitive performance on multi-object tasks, object detection and instance segmentation of COCO 2017 as follows. We evaluate our method on ResNet50 (pretrained for 200 epochs).
> >
> >   |           Method            | AP$^{bb}$ | AP$^{mk}$ |
> >   | :-------------------------: | :-------: | :-------: |
> >   | MosRep (ResNet 50, MoCo-v2) |   40.6    |   36.6    |
> >   |     **MOS (ResNet 50)**     |   41.2    |   37.3    |
> >
> >
> >
> > + **The refinement of the writing about method section**
> >
> >   The subscript $i$ is **indeed necessary** for the introduction of **multiple-to-single** and **multiple-to-multiple labels** as **Eq. 8** and **Eq. 9**. More specifically, since the stitched image is matched to other samples in the image batch, the subscript $i$ is indeed necessary for the construction of object correspondences.

---

> > > ### Comment · Reviewer_AZJE · 2023-12-04
> > > **Response to the Rebuttal**
> > >
> > > Thank you for the rebuttal. I think the empirical results of the paper are pretty nice. However, my concerns regarding presentation and motivation have not been fully addressed. Overall, I have raised my rating to 5.

---

### Official Review · Reviewer_Kxdm · 2023-10-31

**Soundness:** 3 good
**Presentation:** 3 good
**Contribution:** 3 good
**Rating:** 8
**Confidence:** 3

**Summary:**

The research paper introduces a novel method for unsupervised multi-object representation learning in computer vision. It addresses two significant challenges in this field: the semantics inconsistency between views and the difficulty in establishing accurate object correspondences. The proposed method tackles these challenges by synthesizing single-object images into multi-object images. This unique approach is demonstrated to be highly effective in experimental evaluations, excelling in image classification and outperforming existing methods in object detection and instance segmentation. The research opens up new possibilities for unsupervised learning in various modalities.

**Strengths:**

-	The paper presents an innovative approach to unsupervised multi-object representation learning, which is an increasingly important area in computer vision.
-	The method's technique for multi-object image stitching through data augmentation, scaling, and tensor operations is efficient and leads to improved representation learning.

**Weaknesses:**

-	While the method excels in multi-object representations, it may not have been extensively tested in scenarios with highly dynamic or cluttered objects.

**Questions:**

-	In table 3, Selfpatch is trained for only 200 epochs and is compared against this work which is trained for 300 epochs? What is the rationale behind this comparison?
-	Have you tested the method in scenarios with highly dynamic or cluttered objects or occluded objects?

---

> ### Author Response · Authors · 2023-11-15
>
> We appreciate your thoughtful comments and resolve your main concerns as follows:
>
> + **Experiments in scenarios with highly dynamic or cluttered objects**
>
>   Thanks for your constructive suggestion.
>
>   We evaluate our pretrained ViT-S/16 model (with FPN architecture) on WIDER FACE dataset for 80 epochs as [R1], where face objects present high degree of variability in scale, pose, expression, occlusion and illumination. The results (AP when IoU=0.5) are listed as follows.
>
>   |     Method      |  Easy AP  | Medium AP |  Hard AP  |   mAP    |
>   | :-------------: | :-------: | :-------: | :-------: | :------: |
>   | RetinaFace [R1] |   96.9%   |   96.2%   |   91.9%   |   52.3   |
>   |   MOS (ours)    | **97.3%** | **96.8%** | **92.8%** | **52.9** |
>
>   Compared to the previous best method RetinaFace, our method achieves consistent performance gain on easy, medium and hard subsets of WIDER FACE dataset.
>
>   [R1] RetinaFace: Single-stage Dense Face Localisation in the Wild. CVPR 2020.
>
> + **The comparative results about SelfPath pretrained for 300 epochs**
>
>   Thanks for pointing out this typo error. The actual number of pretraining epochs for SelfPatch is 300 epochs. We revise this typo error in the revised paper.

---

### Official Review · Reviewer_R2AJ · 2023-11-01

**Soundness:** 2 fair
**Presentation:** 2 fair
**Contribution:** 2 fair
**Rating:** 5
**Confidence:** 5

**Summary:**

Most existing algorithms for self-supervised learning use single object-centric images for contrastive learning. The paper claims they failed to perform well on images with multiple objects. To resolve the problem, the paper proposes to stitch multiple object-centric images to become multi-object images for contrastive learning. More specifically, the learning objective contains singe-to-single, single-to-multi, and multi-to-multi contrastive pairs. The paper shows improved performance on multiple benchmarks including ImageNet-1K, CIFAR, and COCO.

**Strengths:**

1. The paper achieves state-of-the-art results on multiple benchmarks.
1. The idea is simple but effective.

**Weaknesses:**

1. The motivation is unclear. The paper claims that contrastive learning for single object-centric images "suffer inferior performance on
the widespread images with multiple objects", but it doesn't provide enough evidence to support the claim. For example, ImageNet-1K and CIFAR are recognition problems with single objects, why do we need to stitch multiple together? I understand that, for example, even in ImageNet-1K a lot of times images do contain multiple objects. If this is the case, I suggest the paper show the number of objects vs classification performance by applying some existing detectors, which can prove that the performance of existing algorithms degrades on multi-object images. However, somehow it is counter-intuitive because the proposed method may strengthen the minor objects in the images which distracts the recognition of the main object.
1. Similar to the above, detection and segmentation tasks should be the main and right benchmarks to explain the proposed algorithm because they definitely contain multiple objects in a scene. However, the paper still needs to provide more analysis to support the main assumption about multi-objects. For example, it will be very convincing if the paper shows the proposed algorithm can already achieve a better class-agnostic recall of objects on the attention map on COCO before even fine-tuning on it. In addition, I think it's better if it can show the ablation in table 4 with the detection and segmentation task.
1. To me the idea is simple and I totally understand the proposed algorithm, but I think section 3 is overcomplex with math expression. To me math expression is to help explanation not to make it harder to read. One problem might be there are so many notations. I believe there is a simpler way to make it clearer. For example, stitching itself might not be that important. More important is how to learn from stiched images, e.g. proposed three losses. I hope I can see more underlying reasons for learning objectives instead of just showing the formula. For example, I am not quite convinced by the claim of L_s2s, "to alleviate the domain gap between synthesized multi-object images and natural images". Isn't it just the same as the existing algorithms without stitching?
1. The paper uses ViT as the backbone. I am curious about the performance with CNN because to my understanding this assumption should not be related to the architecture.

**Questions:**

1. In table 4, what is the main difference between s2s and existing unsupervised algorithms?

---

> ### Author Response · Authors · 2023-11-15
>
> Thanks for your constructive comments. We address your main concerns as follows:
>
> + **The evidence for the performance degradation on multi-object images**
>
>   Thanks for pointing out the issue.
>
>   The previous research works **[R1, R2]** indeed support that plain contrastive learning methods for single object-centric images suffer inferior performance on the multi-object images.
>
>   For example, the introduction of **[R1]** claims that "for typical contrastive learning models, the pretraining pretext task considers an image holistically in instance discrimination, without explicit spatial modeling over regions. Though it enhances transferability for classification, this practice is less compatible with spatial reasoning tasks, such as object detection". This clarification is also validated in the experiments, such as Figure 1 of **[R1]**.
>
>   Moreover, our experimental results in Table 5 also demonstrate that our method achieves significant performance gain over the baseline pretrained by single object-centric images on both object detection and instance segmentation.
>
>   We will supplement and emphasize this point in the final revision.
>
>
>
>   **[R1]** Instance Localization for Self-supervised Detection Pretraining. CVPR 2021.
>
>   **[R2]** Self-EMD: Self-Supervised Object Detection without ImageNet. Arxiv 2021.
>
>
>
> + **Why applying image stitching strategy on single-object images, such as ImageNet-1K and CIFAR?**
>
>   Although the images from both ImageNet-1K and CIFAR are single object-centric ones, we surprisingly  find that our method consistently achieve very competitive results on image classification task for single object-centric images. Compared to [R1], our method also boosts the performance on image classification, instead of only spatial-sensitive tasks as [R1]. We speculate that  our proposed stitching method introduces more image context information (include other objects) into training targets, thus alleviating the potential representation degradation (the plain methods may only focus on the most discriminative parts but miss other contents)  and improving the generalization of representations.
>
>
>
> + **More analysis about the assumption on multi-object images**
>
>   Thanks for your constructive suggestion.
>
>   We visualize the self-attention maps of our pretrained model on COCO dataset in Figure 11 (**G The Visualization of Self-Attention Map section**) and object correspondences between multi-object images with the same categories in Figure 12 (**H Object Correspondence**  section).
>
>   As shown in Figure 11, our method can well discriminate different objects in images. For example, the baby and teddy bear of the second row image about can be well distinguished from each others, yet the counterpart, iBOT, fails to discriminate them.
>
>   In Figure 12, our method achieves rough matching between different multi-object images with the same categories. In the first image pair (both images contains dog and person), the correspondences successfully point out the semantic matching for the objects in the images.
>
>   The above visualization results indeed demonstrate the effectiveness of our method on multi-object image representations.
>
> + **More ablation study on detection and segmentation tasks**
>
>   We supplement ablation results on object detection and instance segmentation tasks in Table 4 as follows.
>
>   | $L_{s2s}$ | $L_{m2s}$ | $L_{m2m}$ | COCO AP$^{bb}$ | COCO AP$^{mk}$ |
>   | :-------: | :-------: | :-------: | :------------: | :------------: |
>   |     ✓     |    --     |    --     |      42.8      |      38.9      |
>   |    --     |     ✓     |    --     |      43.1      |      38.8      |
>   |    --     |    --     |     ✓     |       NA       |       NA       |
>   |    --     |     ✓     |     ✓     |      43.6      |      39.2      |
>   |     ✓     |     ✓     |    --     |      44.3      |      39.7      |
>   |     ✓     |     ✓     |     ✓     |    **45.6**    |    **40.6**    |
>
>   When the model is pretrained by all three losses $L_{s2s}$, $L_{m2s}$ and $L_{m2m}$, our method achieves the best results on both object detection and instance segmentation of COCO dataset. It indeed supports the effectiveness of the above three loss items. We supplement these results in the revised paper.

---

> ### Author Response · Authors · 2023-11-15
>
> + **More explanations about $L_{s2s}$**
>
>   The loss $L_{s2s}$ is just the normalized temperature-scaled cross entropy loss, which is widely adopted by the plain contrastive learning methods.
>
>   We explain the design of the contrastive losses $L_{s2s}$ as follows.
>
>   1. The patch boundaries of the stitched images adopted in our method are discontinuous, thus inevitably producing artificiality, which are different from the natural images. We call it domain gap between the stitched images and natural images.
>   2. If we trained the model with only the stitched images, such as $L_{m2s}$ or $L_{m2m}$, the data for training and the ones for testing would be significantly different. Hence, the model would degrade the adaption  of representations on natural images. The results in Table 4 support this clarification, where the training without $L_{s2s}$ obviously degrades the performance on downstream tasks.
>   3. The application of $L_{s2s}$ can introduce natural images during pretraining, thus improving the representation adaption on natural images. It's also supported by the ablation results in Table 4.
>
> + **The performance on CNN**
>
>   As clarified in Introduction section, our method works better on vision transformer architectures. For CNNs, the artificiality produced by the boundary of image stitching may partially affect the representation performance. We evaluate our method on CNN architecture (ResNet50 pretrained for 200 epochs) as follows.
>
>   |       Method        | AP$^{\rm bb}$ | AP$_{50}^{\rm bb}$ | AP$_{75}^{\rm bb}$ | AP$^{\rm mk}$ | AP$_{50}^{\rm mk}$ | AP$_{75}^{\rm mk}$ |
>   | :-----------------: | :-----------: | :----------------: | :----------------: | :-----------: | :----------------: | :----------------: |
>   |  DetCo (ResNet 50)  |     40.1      |        61.0        |        43.9        |     36.4      |        58.0        |        38.9        |
>   | **MOS (ResNet 50)** |     41.2      |        62.4        |        44.7        |     37.3      |        59.2        |        39.5        |
>   | **MOS (ViT-S/16)**  |   **45.6**    |      **65.9**      |      **48.9**      |   **40.6**    |      **62.8**      |      **42.6**      |
>
>   The results demonstrate that our method outperforms the previous state-of-the-art on CNN: DetCo, but worse than the ViT counterpart. It's a promising line to eliminate the gap caused by stitching artificiality. We plan to explore this topic in future work.

---

### Author Response · Authors · 2023-11-15

Dear ACs and Reviewers,

We appreciate your effort to handle our submission. We hope that you could pay more attention to the value of our paper. We believe that our method **addresses an important issue** in multi-object contrastive learning: the construction of accurate object correspondences.

Extensive experiments **indeed support the effectiveness of our method**, e.g., 1.7% linear accuracy improvement on ImageNet-1K, 10.9% linear accuracy improvement on CIFAR100 and 1.3% box AP improvement on COCO, over the previous SOTA. For reproducibility, we also **provide the source code and the trained weights** in the supplementary material.

We believe that our work is able to **make a significant contribution** to the development of unsupervised learning.

We look forward to hearing the decision.

Thanks!

---

### Meta-Review · Area_Chair_HET6 · 2023-12-02

**Metareview:**

The paper introduces a method for synthesizing (stitching) a multi-object image from single-object-centric images, aiming to enhance visual representations in contrastive self-supervised learning. The experimental results demonstrate significant improvements over the methods discussed in the paper across various downstream tasks.

Out of four reviewers, one (Reviewer #Kxdm) provided a highly positive review. Reviewer #Kxdm believes that the method is simple yet effective, supported by strong experimental results. They did, however, highlight a weakness regarding the testing of the proposed method on highly dynamic and cluttered scenes, which the authors addressed through additional experiments. Reviewer #Kxdm did not respond to the authors' rebuttal.

Reviewer #R2AJ provided a slightly negative review and believes that the idea is simple yet effective. The reviewer highlighted weaknesses, such as the lack of motivation for using multi-object stitched images (which the authors addressed in the revised paper), the need for tests on class-agnostic saliency maps (addressed by the authors with samples of saliency maps), the necessity for tests on a different backbone (addressed by additional experiments conducted by the authors), and concerns about the clarity of writing. Reviewer #R2AJ did not respond to the authors' rebuttal.

Two out of four reviewers, namely Reviewer #AZJE and Reviewer #gfC6, submitted reviews that do not favor the paper. Both reviewers share the belief that similar concepts have already been explored, and they noted that recent related works were overlooked by the authors (additional references and experiments were included by the authors). Specifically, Reviewer #AZJE highlights "Mosaic Representation Learning for Self-supervised Visual Pre-training. ICLR'23" as a highly relevant paper, a point I agree with. Both reviewers expressed concerns regarding the clarity of the writing, particularly with regard to the complex mathematical notations used in the paper. R. Reviewer #AZJE was not satisfied with the rebuttal and didn't change the rating for the paper. Reviewer #gfC6 did not respond to the authors' rebuttal

Although the authors put a lot of effort into the paper, given the major concerns regarding novelty and the clarity of the writing, which I agree with, I recommend rejecting the paper.

**Justification For Why Not Higher Score:**

The authors overlooked recent related works, which raises questions about the novelty of the paper.

**Justification For Why Not Lower Score:**

N/A

---

### Decision · Program_Chairs · 2024-01-16

Reject